# A bottom-up reward pathway mediated by somatostatin neurons in the medial septum complex underlying appetitive learning

Li Shen [1,5], Guang-Wei Zhang [1,5], Can Tao[1], Michelle B. Seo[1,2], Nicole K. Zhang[1], Junxiang J. Huang [1,3], Li I. Zhang [1,4 ✉] & Huizhong W. Tao [1,4 ✉]

Valence detection and processing are essential for the survival of animals and their life quality in complex environments. Neural circuits underlying the transformation of external sensory signals into positive valence coding to generate appropriate behavioral responses remain not well-studied. Here, we report that somatostatin (SOM) subtype of GABAergic neurons in the mouse medial septum complex (MS), but not parvalbumin subtype or glutamatergic neurons, specifically encode reward signals and positive valence. Through an ascending pathway from the nucleus of solitary tract and then parabrachial nucleus, the MS SOM neurons receive rewarding taste signals and suppress the lateral habenula. They contribute essentially to appetitive associative learning via their projections to the lateral habenula: learning enhances their responses to reward-predictive sensory cues, and suppressing their responses to either conditioned or unconditioned stimulus impairs acquisition of reward learning. Thus, MS serves as a critical hub for transforming bottom-up sensory signals to mediate appetitive behaviors.

[1] Zilkha Neurogenetic Institute, Keck School of Medicine, University of Southern California, 1501 San Pablo Street, Los Angeles, CA 90033, USA. [2] Neuroscience Graduate Program, University of Southern California, Los Angeles, CA 90089, USA. [3] Graduate Programs in Biomedical and Biological Sciences, Keck School of Medicine, University of Southern California, Los Angeles, CA 90033, USA. [4] Department of Physiology & Neuroscience, Keck School of Medicine, University of Southern California, 1501 San Pablo Street, Los Angeles, CA 90033, USA. [5] These authors contributed equally: Li Shen, Guang-Wei Zhang. ✉email: liizhang@usc.edu; htao@usc.edu

Detecting the positive or negative valence of external sensory cues and transforming the valence value into appropriate behavioral reactions are essential for animals' survival and well-being in complex and challenging environments. Negative valence signals induce aversion/avoidance behaviors, while positive valence signals result in appetitive/approaching behaviors[1–3]. Although extensive previous studies have been focused on how neural structures in an emotional processing network and their distinct cell groups contribute to the aversion or rewarding type of behaviors[1,2,4–6], how ascending sensory signals are transmitted to these structures is poorly studied in general. There has been increasing evidence suggesting that for each sensory modality there could be a distinct ascending neural pathway devoted to valance detection and processing, which is independent of the canonical thalamocortical pathway for generating sensory perception. For example, the medial septum complex (MS) in the basal forebrain receives bottom-up aversive sensory information of multiple modalities, such as auditory and somatosensory signals, from pontine nuclei and mediates related sensory-induced avoidance behaviors[7,8]. In comparison, analogous bottom-up sensory pathways to process and transform positive valence signals into appetitive behaviors have been less well understood[2,9].

For an efficient and economic architecture of emotional processing networks, multisensory signals of the same valence from distinct ascending pathways may arrive and converge at the same critical processing node, following which a common effector (motor-related) pathway is modulated to elicit an aversive or approaching behavior. Within a well-studied emotional processing network[2,9], the lateral habenula (LHb) has been identified as one of the critical structures to mediate reward/aversion[10–12]. It modulates both dopamine (DA) and serotonin (5-HT) neuromodulatory systems, which are considered to be central for reward/aversion decision making[2,9]. A distinct feature of LHb is that it contains predominantly excitatory glutamatergic neurons[13]. These neurons are found to respond opposingly to sensory stimuli with positive and negative valences, i.e. being activated by aversive signals and suppressed by reward signals[10,14]. Several input structures upstream of LHb, such as the ventral pallidum and lateral preoptic area, project both excitatory and inhibitory axons to LHb[5,15,16]. The excitatory projections are found to carry negative valence, which activates LHb to negatively influence the DA and 5-HT systems partly through the GABAergic rostromedial tegmental nucleus[17,18], resulting in anti-rewarding effects or aversion[9,11]. On the other hand, the GABAergic projections carry positive valence, which reduces LHb activity and results in rewarding emotional effects[5,15]. Thus, by receiving convergent but opponent positive and negative valence signals, LHb may serve as a hub to integrate valence information from a variety of sources.

Both top-down and bottom-up information is relayed to LHb. Among its input sources, MS has been found to transmit bottom-up aversive sensory signals to LHb via its glutamatergic neurons, resulting in negative emotion related behaviors[6,8]. Although GABAergic neurons in MS are also found to innervate LHb neurons[7], their functional contributions under different behavioral contexts remain largely unclear. MS contains different subtypes of GABAergic neurons, including parvalbumin (PV) and somatostatin (SOM) expressing neurons, which may receive distinct inputs and form intra-nuclear connections with glutamatergic neurons[19,20]. One possibility is that some specific types of MS GABAergic neurons may receive and transform ascending reward-related information via distinct sensory pathways to mediate appetitive behaviors.

In the present study, we tested this idea by combining behavioral assays, in vivo and in vitro electrophysiology as well as opto/chemogenetic and pharmacological manipulations. We found that MS GABAergic neurons encode positive motivational valence, respond preferentially to rewarding gustatory signals relayed from the parabrachial nucleus (PBN) in the pons, primarily through the SOM subtype of these inhibitory neurons. The SOM neurons in MS acquire responses to reward-predicative cues through cue-reward associative learning and mediate the acquisition of cue-reward association via their projections to LHb. Together, these results highlight an essential role of MS in the formation of reward associative memory.

## Results

**MS GABAergic neurons encode reward signals.** To test whether MS GABAergic neurons can encode positive valence, we utilized a pan-GABAergic Cre diver line, Vgat-Cre, and injected adeno-associated virus (AAV) encoding Cre-dependent channelrhodopsin2 (ChR2)[21] (or GFP alone as control) to activate GABAergic neurons in the medial septum complex (shortened as MS in this study). Post-hoc histology revealed that the infected area included the medial septum and the medial part of diagonal band nucleus (NDB) (Supplementary Fig. 1a). For photostimulation, 470 nm blue LED light (at 20 Hz, 5 ms pulse duration) was delivered to MS via an implanted optical fiber (see "Methods"). We first adopted a two-choice self-stimulation paradigm to assess the reinforcing value associated with the activation. LED stimulation was triggered whenever the animal nose-poked the designated LED-on port, whereas nose-poking the other port did not trigger any photostimulation (Fig. 1a). During a 30-min test session, we observed a much greater number of pokes into the LED-associated port than the control port, while in GFP-expressing control mice the number of pokes was not significantly different between the two ports (Fig. 1b, c). This result demonstrates that acute optogenetic activation of MS GABAergic neurons reinforces self-stimulation behavior, which is an important hallmark for reward-encoding neurons[22–24]. Moreover, in a two-chamber light-dark box test[25], ChR2-expressing animals spent significantly more time in the light chamber, which was paired with LED stimulation, than GFP control mice (Fig. 1d, e), consistent with the notion that activation of these neurons is rewarding and counteracts the tendency of mice to avoid bright areas[25]. In a two-chamber real-time place preference (RTPP) test, LED stimulation was applied whenever the animal entered and stayed in the designated stimulation (LED-on) chamber. Similar to GAD2-Cre animals[7,26], we found that the ChR2-expressing Vgat-Cre animals spent significantly more time in the stimulation chamber compared to GFP-expressing controls (Fig. 1f, g), confirming that activation of MS GABAergic neurons as a population drives place preference and thus is rewarding. Together, these results indicate that MS GABAergic neurons encode positive motivational valence.

Using optrode recording (see "Methods"), we next directly examined whether MS GABAergic neurons could respond to reward signals in awake head-fixed mice licking sucrose water (5% w/v, 10 μL per trial) (Fig. 2a). The GABAergic neurons were tagged either by crossing the Vgat-Cre with Ai27 (Cre-dependent ChR2) mice or by injecting AAV-floxed-ChR2 into MS of Vgat-Cre animals. For ChR2-tagged (GABAergic) neurons, an increase of firing rate following the licking onset was observed in a majority of the neurons recorded (79%, 59 out of 75 units), while 13% of the population (10/75) showed no change in firing rate and 8% (6/75) showed a decrease in firing rate (Fig. 2b, c). In the untagged population which presumably contained largely non-GABAergic neurons, only 24% (23/94) showed excitatory responses to sucrose and the majority showed no responses (63%, 59/94) (Fig. 2b, c). In all the recorded neurons that showed

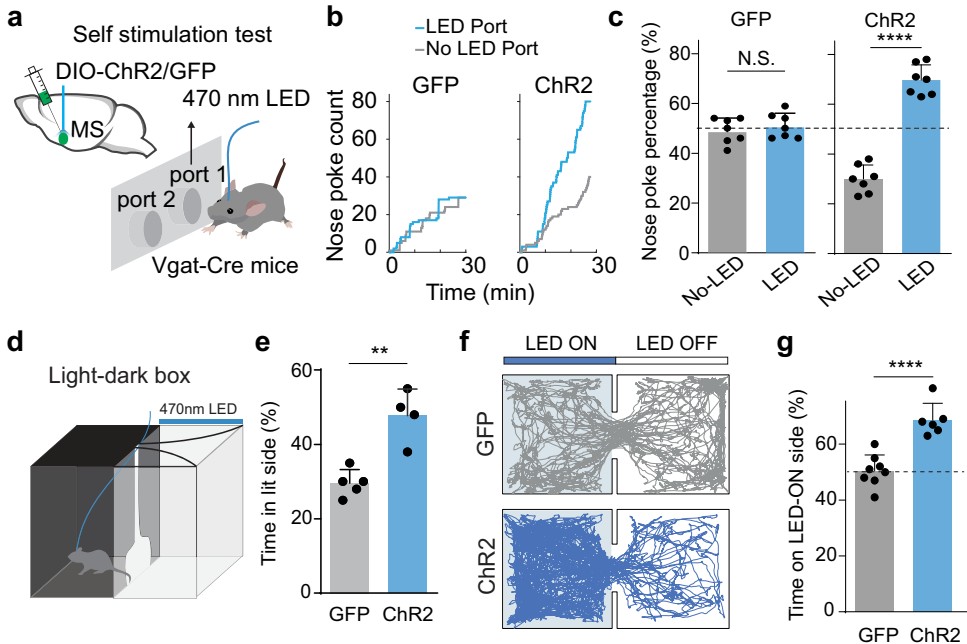

**Fig. 1 Activation of MS GABAergic neurons is reinforcing and drives positive valence. a** Nose-poke self-stimulation test. Animal's poking of port 1 (randomly assigned) triggered LED stimulation of MS GABAergic neurons via an implanted optical fiber. Vgat-Cre mice were injected with AAV-DIO-ChR2 or AAV-DIO-GFP in MS. **b** Cumulative counts of nose pokes of the port receiving LED stimulation (blue) and the one without LED stimulation (gray) over 30-min test window for two example mice with ChR2 or GFP expressed in MS. **c** Percentage count of nose-poking of each port. N.S., not significant, paired $t$ test, $n = 7$ mice. GFP, $p = 0.6259$; ChR2, ****$p < 0.0001$, two-sided paired $t$ test, $n = 7$ mice. **d** Illustration of light-dark box test. LED stimulation was applied whenever the animal was in the light chamber. **e** Percentage time spent in the light chamber for GFP control ($n = 5$) and ChR2-expressing ($n = 4$) mice. **$p = 0.0016$, two-sided $t$ test. **f** Movement tracking traces for a GFP control (gray) and a ChR2-expressing mouse (blue) in the RTPP test. LED stimulation was applied whenever the animal was in the LED-On chamber. **g** Percentage time spent in the LED-ON chamber for GFP control ($n = 8$) and ChR2-expressing ($n = 6$) mice. ****$p < 0.0001$, two-sided $t$ test. All error bars in this figure indicate s.d. Source data are provided as a Source Data file.

excitatory responses to sucrose, 72% were ChR2-tagged (Fig. 2d). These results suggest that the GABAergic population was preferentially activated by sucrose licking. The firing rate increase in the GABAergic neurons was due to the consumption of sucrose water (i.e. reward) but not to the licking behavior per se, as it was not observed in trials when the animal licked but no sucrose water was delivered (Fig. 2g). We performed similar recordings in animals passively receiving quinine (5 mM, 10 μL), a bitter tastant, via an intraoral cheek fistula into the oral cavity (see "Methods"). We did not observe significant changes in firing rate (Fig. 2g). In contrast to the GABAergic neurons, MS glutamatergic neurons, as tagged in Vglut2-Cre mice, did not respond to sucrose consumption but were activated by quinine (Fig. 2e–g), consistent with the previous finding that activity of these neurons encodes aversive emotional outcome[7]. Together, these results indicate that MS GABAergic neurons as a population are preferentially activated by reward signals in contrary to their glutamatergic counterpart. More importantly, their responses to sucrose became stronger with increasing volumes (3, 6, 10 μL) consumed (Fig. 2h, i), suggesting that the activity level can encode reward value.

**PBN provides bottom-up rewarding gustatory input to MS.** We wondered what input source relayed the rewarding (sweet) gustatory input to MS. It is known from work in rodents that taste information from peripheral receptors is relayed centrally to the nucleus of the solitary tract (NST) and then to the PBN in the pons[27,28]. A number of taste-related projections arise from PBN, with some carrying taste information to the gustatory thalamus (VPMpc) before reaching the gustatory cortex[29,30]. To identify the input source of MS GABAergic neurons, we performed cell-type

specific monosynaptic input tracing with pseudo-typed rabies virus[31,32] (Fig. 3a). We observed numerous retrogradely labeled cells in PBN, but not in NST, VPMpc or gustatory cortex (Fig. 3b). Consistent with previous work[33], in vivo extracellular recording revealed that PBN neurons responded to sucrose by increasing their firing rates (Fig. 3c, d). Anterograde tracing by injecting AAV-floxed-GFP in PBN of Vglut2-Cre mice confirmed that PBN glutamatergic neurons projected their axons to MS (Fig. 3e). Slice whole-cell recordings from MS neurons in Vglut2-Cre mice injected with AAV-floxed-ChR2 in PBN further confirmed monosynaptic connectivity between PBN glutamatergic axons and MS neurons (Fig. 3f). Application of CNQX completely blocked the light-evoked synaptic current recorded at −70 mV, further confirming that the current was mediated by AMPA receptors (Supplementary Fig. 2). Together, these results demonstrate that MS receives gustatory input directly from PBN.

To further test in vivo whether PBN relays rewarding taste signals to MS GABAergic neurons, we silenced PBN bilaterally with muscimol during optrode recording (Fig. 3g). The recording was performed before (as baseline) and 15 min after muscimol infusion. The muscimol application resulted in a profound reduction (by ~80% on average) in the response of MS GABAergic neurons to passively received sucrose (Fig. 3h–j). Likewise, bilateral silencing of NST (Fig. 3k) nearly completely blocked the sucrose response of MS GABAergic neurons (Fig. 3l–n). The reduction of sucrose responses was not observed in control animals that received vehicle infusion in either of these structures (Supplementary Fig. 3).

Previously, sucrose-induced suppressive responses have been observed in LHb neurons[14]. Indeed, in vivo recordings revealed a sucrose-induced suppressive response in LHb neurons, which was largely blocked by silencing MS (Fig. 3o–r). Therefore, the MS

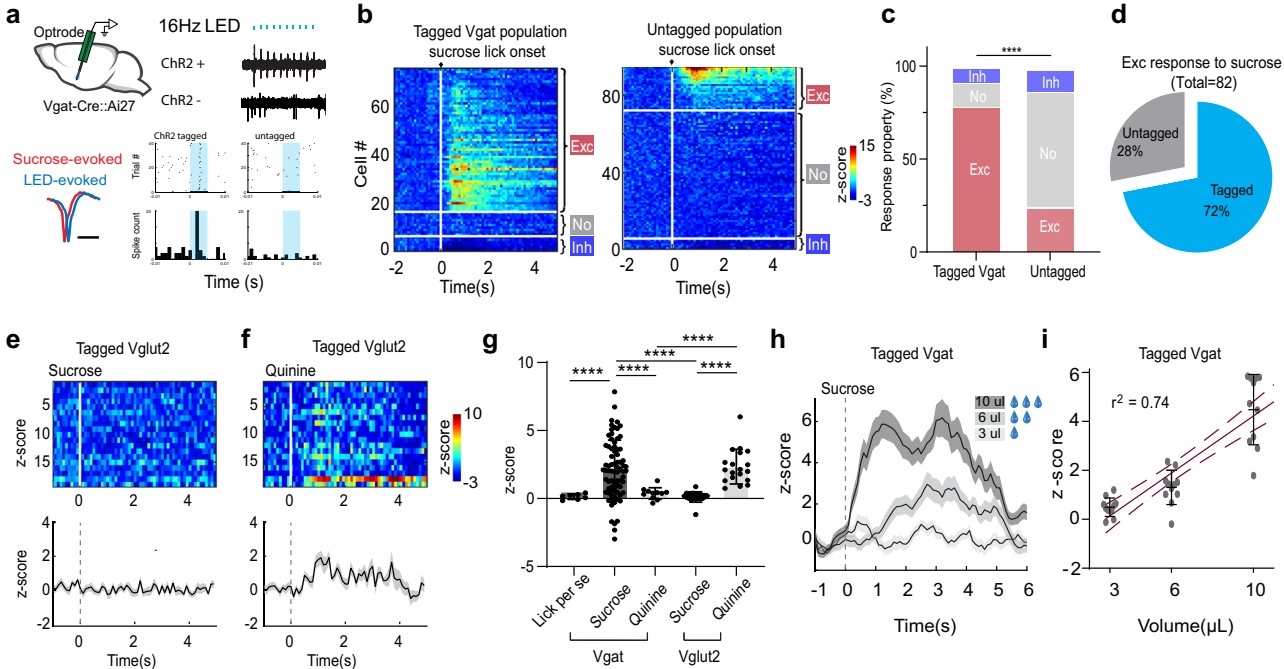

**Fig. 2 MS GABAergic neurons encode reward signals. a** Optrode recording from MS GABAergic neurons while sucrose water (5% w/v) was delivered and animal licked. Upper right, LED pulses (at 16 Hz, blue dots) evoked spikes in an example tagged GABAergic neuron and an untagged neuron. Lower left, average spike waveforms for LED-evoked (blue) and sucrose-evoked (red) spikes in the same tagged units. Note that the waveforms are slightly offset for comparison. Scale bar, 0.5 ms. Lower right, raster (upper) and peri-stimulus spike time histogram (PSTH, lower) plots of spikes of the example neurons. Shaded blue indicates the duration of the LED pulse. **b** Heatmap plot of Z-score for all recorded tagged GABAergic units (ChR2+, $n = 75$), and untagged units (ChR2−, $n = 94$). They were ranked and divided into three subgroups: excitatory response, no response, suppressive response. **c** Proportion of cells in each subgroup for tagged and untagged populations, ****$p < 0.0001$, $\kappa^2$ Test, two-sided. **d** Percentages of tagged and untagged neurons in the recorded population activated by sucrose. **e**, **f** Heatmap plot of peri-stimulus Z-score (upper) for all the recorded MS Vglut2+ units ($n = 19$) and the corresponding population average (lower) in response to sucrose (**e**) or 5 mM quinine (**f**) passively received. Gray shade indicates s.e.m. **g** Comparison of the mean Z-score within a 0–2 s window after the onset of licking per se (without liquid delivered, $n = 7$), or in response to sucrose or quinine for Vgat+ neurons ($n = 75$ and 11, respectively) and to sucrose/quinine for Vglut2+ neurons ($n = 19$). ****$p < 0.0001$, one-way ANOVA with post-hoc Tukey's test, two-sided. **h** Population average of peri-stimulus Z-score for Vgat+ neurons ($n = 11$) in response to sucrose of different volumes (3 µL, 6 µL, or 10 µL). Gray shade indicates s.e.m. **i** Mean Z-score plotted against sucrose volume, $n = 11$ cells. Dark red solid line shows the best-fit linear regression line. Dashed lines, 95% confidence intervals. Source data are provided as a Source Data file.

GABAergic projection to LHb[7] mediates the reward-related suppression in LHb. Consistent with the changes of neuronal responses, silencing PBN, NST or MS reduced sucrose-induced licking behavior (Supplementary Fig. 4), suggesting reduced appetitive motivation. Together, our data reveal a bottom-up ascending pathway for relaying appetitive taste information: from NST to PBN and then to MS GABAergic neurons, which then inhibit LHb neurons.

**MS GABAergic neurons are required for reward associative learning.** Since MS GABAergic neurons encode reward signals, they may play a role in reward associative learning. To test this idea, we employed a spatial reward learning paradigm (see "Methods"). Water-restricted mice were trained to perform reward-seeking in a hole-board[34], with a target hole hiding sucrose water (Fig. 4a). Over four days' training, latency for the animal to locate the reward (Fig. 4b, left) became gradually shortened (Fig. 4c, black), indicating successful association of a spatial location with reward. We then chemogenetically suppressed MS GABAergic neurons by expressing the inhibitory designer receptor exclusively activated by designer drugs (DREADD) receptor, hM4Di[35], and administrating the DREADD agonist, CNO, daily before the training. The effectiveness of CNO was verified using in vitro slice recording (Supplementary Fig. 5). The chemogenetic silencing greatly impaired the spatial reward learning (Fig. 4b, right), as shown by the much longer latency to

locate the reward on day 4 as compared to mCherry control animals (Fig. 4c). This result suggests that MS GABAergic neurons are required for the reward associative learning.

Next, we used a cue-reward associative learning paradigm[36,37], where the animal learned to lick for reward (sucrose water) upon hearing a sound (see "Methods"). As MS GABAergic neurons can encode reward value, they may provide the unconditioned stimulus (US) signal for this type of associative learning. To test this idea, we optogenetically silenced MS GABAergic neurons by injecting AAV encoding Cre-dependent archaerhodopsin-3 (AAV-DIO-ArchT, or AAV-DIO-GFP as control) in Vgat-Cre mice (Fig. 4d). Green LED light illumination was applied during the sucrose consumption (US) window to silence US-related activity of the neurons, over 5 days' training of coupling the conditioned stimulus (CS, 2-s tone) and US (10 µL sucrose) (Fig. 4e). On the 6th (test) day, no LED light illumination was applied. Behavioral responses to both CS and US were monitored. We found that over the course of conditioning (day 1- day 5) the anticipatory licking rate (i.e. licks during the time window between the onsets of CS and US, with baseline subtracted) was gradually increased in GFP control mice (Fig. 4e, right). This increase in anticipatory licking was also observed on the test day, indicating that the animal had learned to associate the sound cue with reward. The increase in anticipatory licking was however largely prevented in ArchT-expressing animals (Fig. 4f, g). Meanwhile, licking during the sucrose

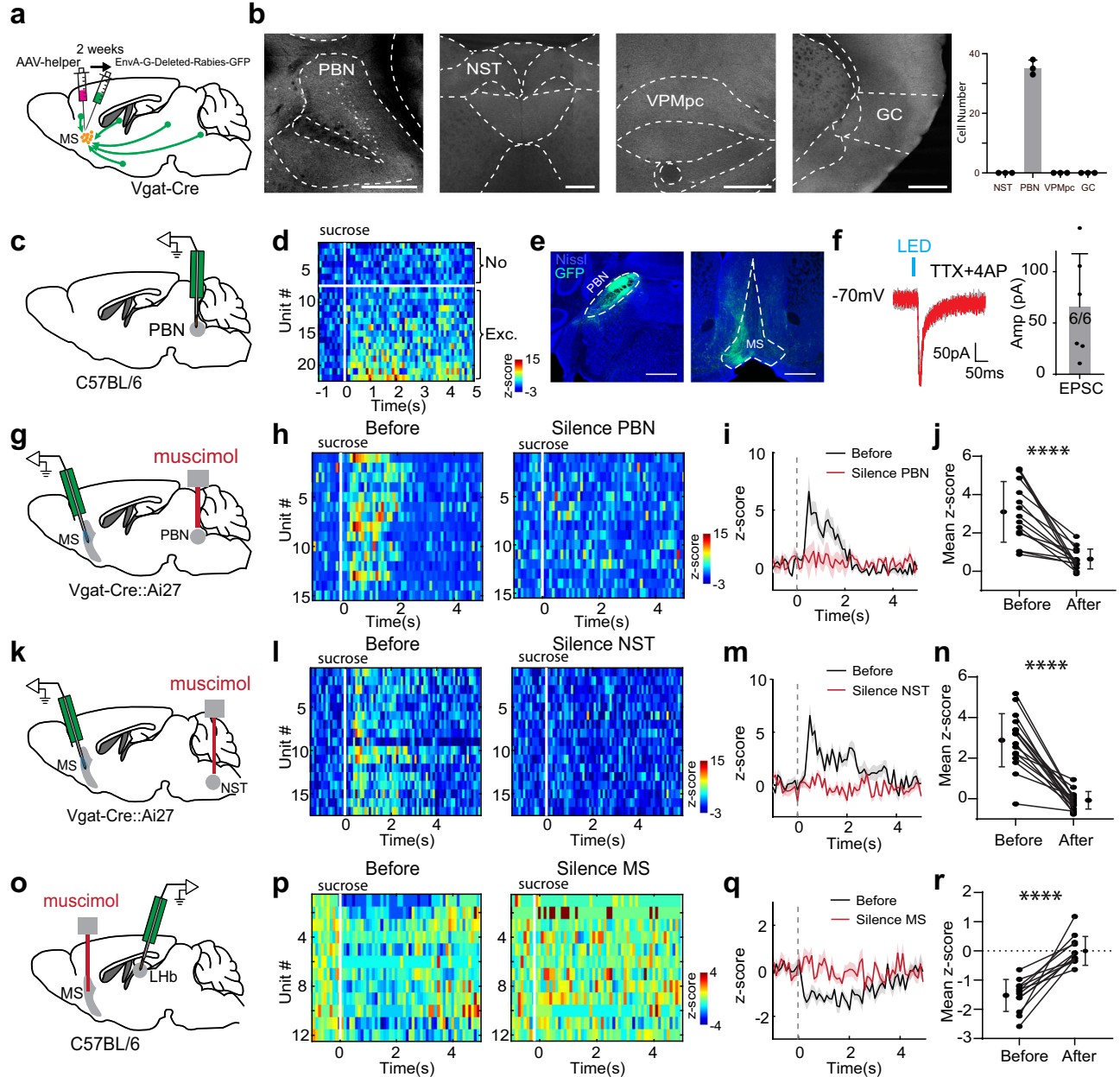

**Fig. 3 A bottom-up sensory pathway to relay rewarding gustatory signals via MS. a** Injection strategy for the retrograde tracing of monosynaptic inputs to MS GABAergic neurons. **b** Retrogradely labeled neurons in PBN, but no labeling in NST, gustatory thalamus (VPMpc) or gustatory cortex (GC). Scale bar, 500 μm. Right, average number of labeled neurons in each area ($n = 3$ experiments). **c** Multichannel recording in PBN. **d** Heatmap plot of Z-score for PBN neuron responses to sucrose ($n = 22$ cells). 68% (15/22) of recorded neurons displayed a significant excitatory response to sucrose. **e** Labeling of PBN Vglut2+ neurons by injecting AAV-DIO-GFP in Vglut2-Cre mice ($n = 3$ experiments). Left, GFP expression at the injection site. Right, GFP-labeled PBN axons in MS. Scale bar, 500 μm. **f** Left, LED-evoked EPSCs recorded in an example MS neuron in the presence of TTX and 4AP. ChR2 was expressed in PBN Vglut2+ neurons. Gray, all recorded EPSCs. Red, average trace. Scale bar, 50 pA, 50 ms. Right, average EPSC amplitudes recorded in 6 MS neurons. All error bars in this figure indicate s.d. **g** Optrode recording from MS GABAergic neurons while pharmacologically silencing PBN bilaterally with muscimol. **h** Heatmap plot of Z-score for sucrose responses of MS GABAergic neurons before and after silencing PBN. **i** Population average. All shades indicate s.e.m. **j** Mean Z-score before and after silencing PBN ($n = 15$ cells). Data points for the same neuron are connected with a line. ****$p < 0.0001$, two-sided paired $t$ test. **k–n** Similar to (**g–j**), but for silencing NST ($n = 17$ MS neurons). ****$p < 0.0001$, two-sided paired $t$ test. **o** Recording from LHb neurons while pharmacologically silencing MS with muscimol. **p** Heatmap plot of Z-score for sucrose responses of LHb neurons before and after silencing MS. **q** Population average. **r** Mean Z-score before and after silencing MS ($n = 12$). ****$p < 0.0001$, two-sided paired $t$ test. Source data are provided as a Source Data file.

consumption window was reduced by silencing MS GABAergic neurons as compared to GFP control animals (Supplementary Fig. 6a). These results indicate that the US responses of MS GABAergic neurons are necessary for forming the cue-reward association.

**Learning strengthens responses of MS GABAergic neurons to reward-predictive cues.** Neurons involved in reward learning may change their responses during conditioning[14,36,37]. To test how the responses of MS GABAergic neurons are shaped by learning, we performed optrode recording before, during, or after conditioning

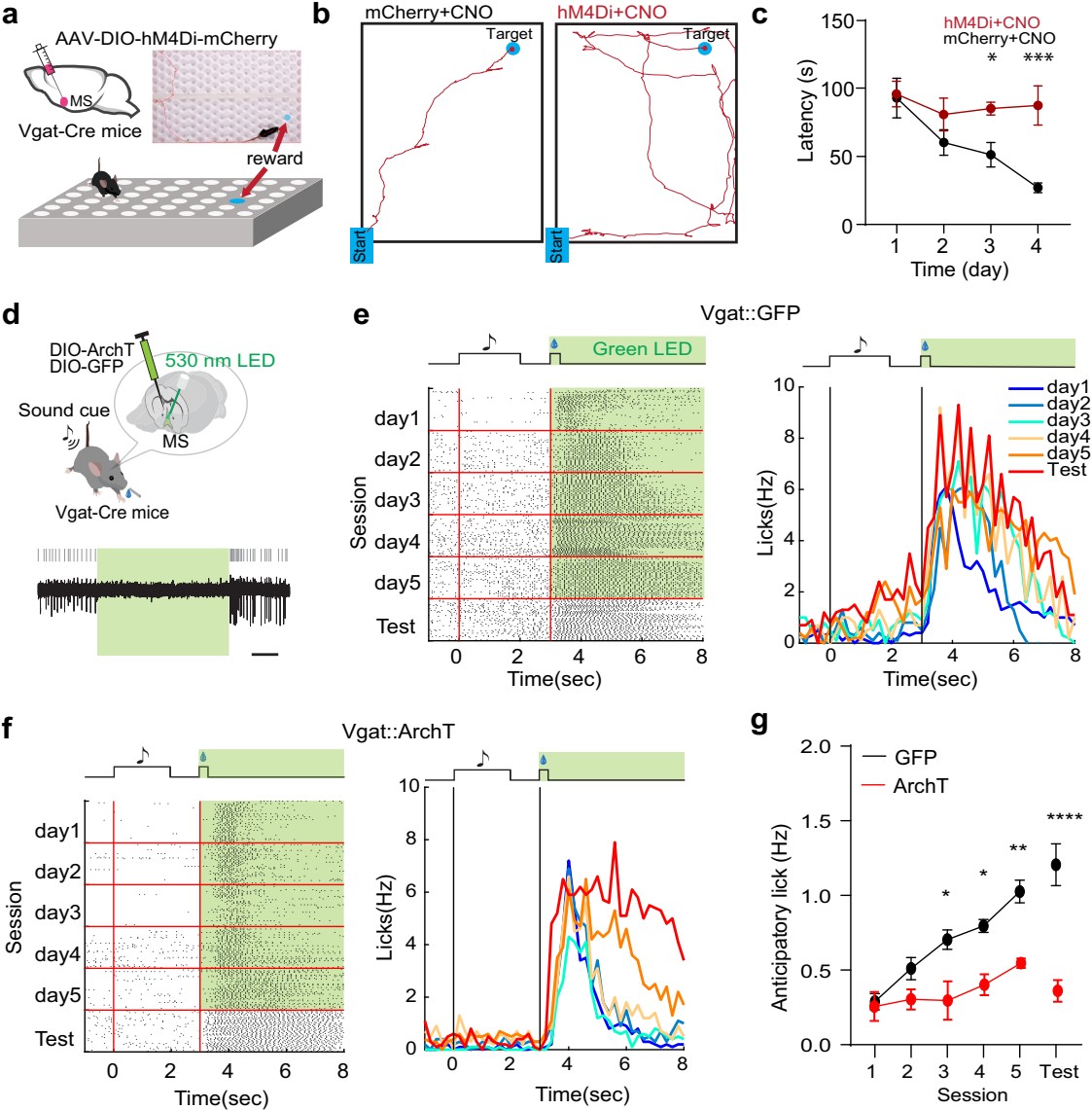

**Fig. 4 MS GABAergic neurons are required for reward associative learning. a** Hole-board test while chemogenetically silencing MS GABAergic neurons. Top inset, photograph with superimposed movement track of an example animal locating the reward. **b** Movement tracks for an example mCherry control animal injected with CNO (left) and a hM4Di-expressing animal injected with CNO (right) tested on day 4. **c** Average latencies for reward localization over training days. Day1, $p = 0.8586$, D2, $p = 0.1863$, D3, $*p = 0.0264$, D4, $***p = 0.0008$, two-way ANOVA, post-hoc Fisher's LSD test, two-sided, $n = 3$ mice in each group. Error bars indicate s.e.m. **d** Cue-reward associative learning while optogenetically silencing MS GABAergic neurons. Bottom, slice recording from an ArchT-expressing MS neuron showing suppression of neuronal spikes by green light (530 nm). Scale, 1 s. **e** Raster plot of licking events (left) or peri-event lick rate (right) during training sessions (over 5 days) and in the test session (on day 6) for an example GFP control mouse. Green LED light was applied during the US delivery window (5 s) throughout trials. Licks between onsets of CS (sound cue) and US (sucrose) are defined as anticipatory licks. **f** Similar to (**e**), but for an ArchT-expressing mouse. **g** Average anticipatory lick rates across training sessions and in the test session for GFP control ($n = 4$) and ArchT ($n = 3$) groups. D1, $p > 0.9999$, D2, $p = 0.6017$, D3, $*p = 0.0132$, D4, $*p = 0.0188$, D5, $**p = 0.0032$, Test, $****p < 0.0001$, two-way ANOVA, post-hoc Bonferroni test, two-sided. Error bars indicate s.e.m. Source data are provided as a Source Data file.

for two days (Fig. 5a). Before conditioning (i.e. in naïve mice), we did not observe significant responses in these neurons to the tone that would be used as the CS (Fig. 5b, f). During the first day of conditioning, the neurons exhibited at best a weak response to the CS while a strong response to sucrose (Fig. 5c, f). On the second day of conditioning, the response to the CS became much stronger (Fig. 5d, f). Following two days' conditioning, the robust response to CS persisted into the third day, when only CS was presented without US (Fig. 5e, f). On average, the response to CS was increased by 5.8-fold compared to the first day of conditioning (Fig. 5f). These results demonstrate learning-induced plasticity of MS GABAergic neuron responses to reward-predictive cues.

**SOM but not PV neurons account for the reward processing in MS.** In the basal forebrain, GABAergic neurons are known to be functionally diverse[19]. Parvalbumin (PV) and somatostatin (SOM) have been used as markers for different subtypes of GABAergic neurons in both the basal forebrain[19,20,38] and other brain regions[39,40]. We investigated which of these GABAergic cell subtypes played a role in the reward learning by first examining their responses to reward signals. We expressed Cre-dependent ChR2 in MS of PV-Cre or SOM-Cre mice and performed optrode recording. PV or SOM positive units were identified by recorded spikes time-locked to LED light pulses applied (Fig. 6a). Overall, 86.4% of the recorded SOM neurons were activated by sucrose

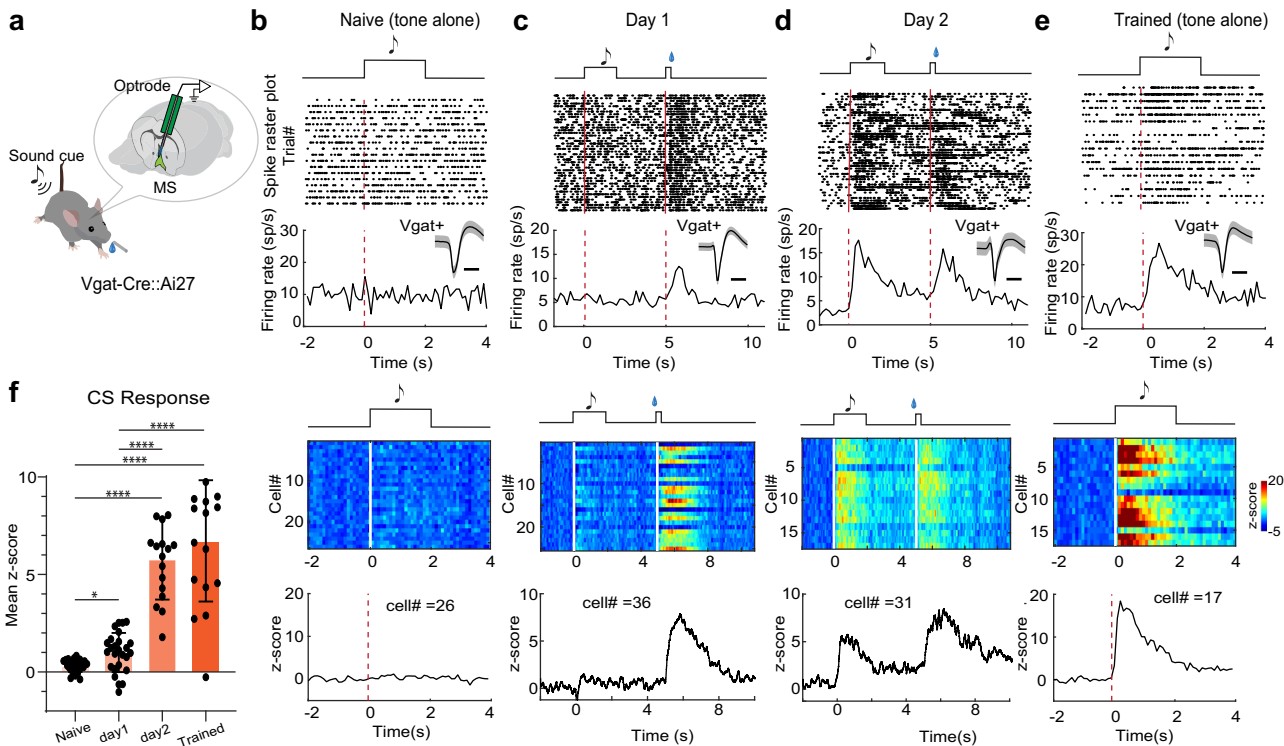

**Fig. 5 Plasticity of MS GABAergic neuron responses to reward-predictive cues. a** Optrode recording from MS GABAergic neurons during conditioning when a sound cue (CS) was presented repetitively before sucrose (US) delivery. **b** Top, raster plot of spikes of an example MS GABAergic neuron in response to tone (upper) and the corresponding peri-stimulus spike time histogram (PSTH) (lower) in a naïve mouse. Inset: average spike waveform of this neuron (shade, s.d.). Scale, 0.5 ms. Red dash line marks the onset of the tone. Bottom, heatmap for responses of all recorded neurons (n = 26, upper) and the population average of Z-score (lower). **c, d** Similar to (**b**), but for mice during two days' training (n = 36 and 31 for day 1 and day 2 respectively, 50 trials per session). Red dash line marks the onset of CS and US, respectively. The heatmap illustrates data from one animal while the lower Z-score plot is the population average across 3 animals. **e** For mice which had completed two day's training (n = 17 cells). **f** Mean Z-score during tone stimulation (0–2 s) in naïve mice, during training (day 1, day 2) and after training (trained). Naïve vs D1, *p = 0.015, D2 vs Trained, p = 0.9046; ****p < 0.0001, one-way ANOVA and post-hoc Tamhane's T2 multiple comparisons test, two-sided. Source data are provided as a Source Data file.

and 13.6% showed no response (Fig. 6b, d, e), whereas 87.0% of the PV neurons showed no response to sucrose and 13.0% were suppressed (Fig. 6c–e). Also consistent with the properties of the general GABAergic population, the SOM neurons responded more strongly to sucrose than just water, as shown by $Ca^{2+}$ signals recorded by in vivo fiber photometry (Fig. 6f, g), indicating that they can encode reward value. By expressing ChR2 in PBN and recording from tdTomato-labeled SOM neurons in MS of slice preparations, we further confirmed that the SOM neurons received excitatory monosynaptic input from PBN (Fig. 6h, i).

In addition, activation of the SOM neurons produced place preference in the RTPP test (Fig. 6j–l), similar to GABAergic neurons, whereas activation of the PV neurons resulted in weak place avoidance (Fig. 6m, n). Together, these results suggest that the SOM neurons encode positive valence and convey reward information. In contrast, the PV neurons encode negative valence and they are unlikely involved in appetitive learning.

To test SOM neurons' involvement in reward learning, using optrode recording we examined spiking activity of these neurons before and during reward associative learning with the same appetitive Pavlovian conditioning paradigm. Before conditioning, the SOM neurons showed no response to the CS tone (Fig. 7a). During the first and second day of conditioning, we observed gradual emergence and strengthening of their responses to the CS tone (Fig. 7b–e), similar to what had been observed for the GABAergic neurons as a whole population. Altogether, these results suggest that the SOM subtype of MS GABAergic neurons likely contributes to the reward associative learning.

**MS SOM neurons are required for reward associative learning.** To directly examine the contribution of SOM neurons to reward associative learning, we expressed Cre-dependent ArchT in MS of SOM-Cre mice. We applied photoinhibition during either the US window (Fig. 8a–d) or the CS plus delay window (Fig. 8e–h). Compared to GFP control animals, suppressing either CS-related or US-related activity of the SOM neurons largely impaired the reward associative learning, as shown by the prevention of increases of anticipatory licking over conditioning sessions and in the test session after the conditioning (Fig. 8d, h). Silencing the SOM neurons during the US window also reduced licking to sucrose as compared to GFP control mice (Supplementary Fig. 6b). Interestingly, in well-trained mice, photoinhibition of the SOM neurons during the CS plus delay window had no effect on anticipatory licking (Fig. 8i–l), suggesting that after the acquisition of associative learning the activity of the SOM neurons is not required for the expression of cue-reward association. However, it is indispensable for the formation of cue-reward association during conditioning. Together, these results indicate that SOM neurons in MS are required for the acquisition of reward associative learning.

**The $MS_{SOM} \rightarrow LHb$ pathway mediates reward associative learning.** The blockade of the suppressive reward response in LHb by silencing MS (Fig. 3o–r) suggested that MS SOM neurons might mediate reward associative learning through LHb. We then anterogradely traced axons from GFP-labeled MS SOM neurons and found robust projections to LHb

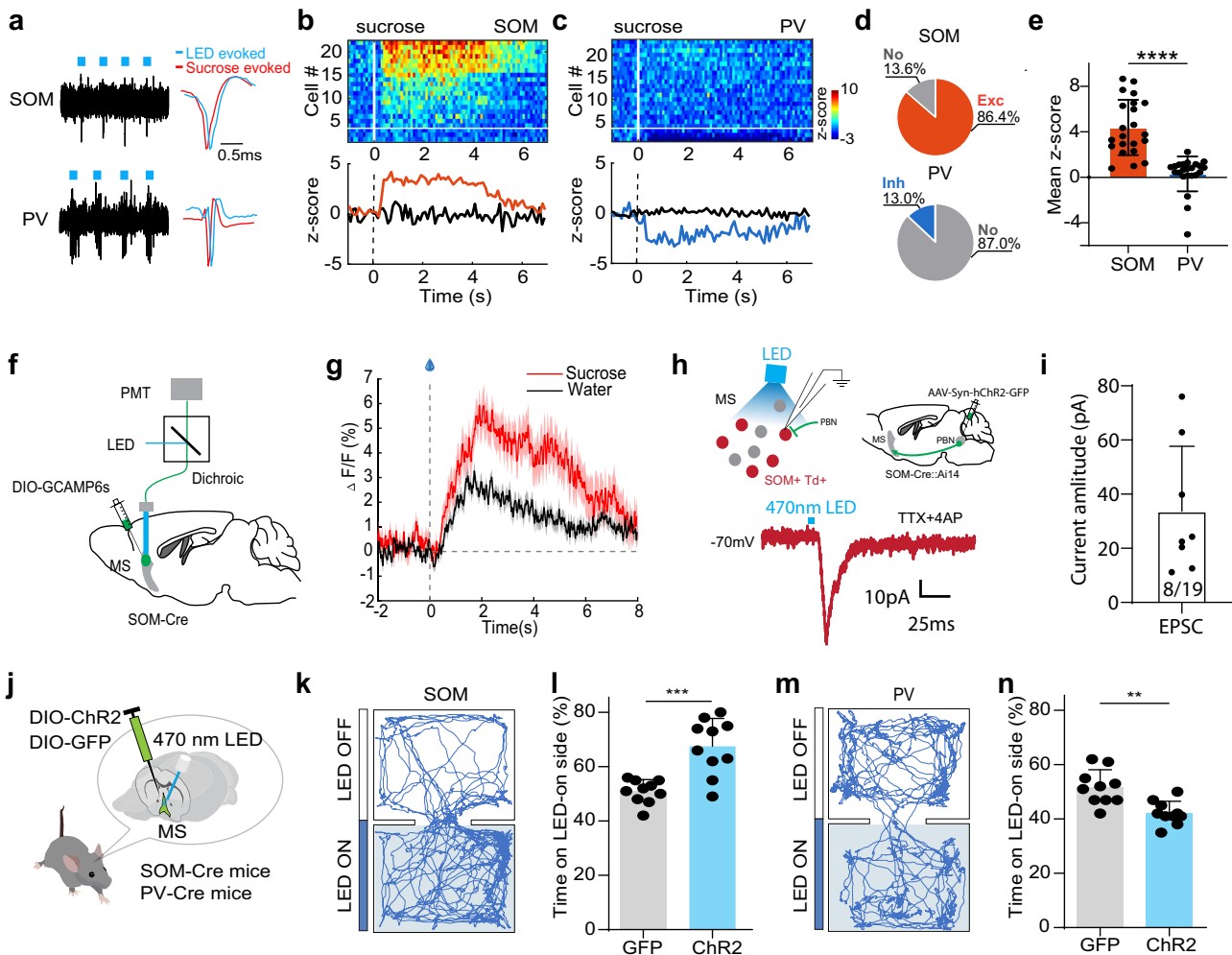

**Fig. 6 SOM but not PV MS neurons encode reward signals. a** Spikes of an example SOM (top) and PV (bottom) MS neuron to pulses of LED light (blue dot) in optrode recording. Right inset, comparison of spike waveforms evoked by LED (blue) and sucrose (red). **b** Top, heatmap plot of peri-stimulus Z-score for responses of SOM neuron to sucrose. 19/22 units showed significant excitatory responses while 3/22 units did not show significant responses (divided by the white horizontal line). Bottom, population average of Z-score for neurons showing excitatory (red) or no (black) responses. **c** PV neuron responses to sucrose. 3/23 units showed significant inhibitory responses while 20/23 units did not show significant responses. **d** Proportion of responsive versus non-responsive neurons in SOM and PV groups. **e** Mean Z-score value for all the recorded SOM (n = 22) and PV (n = 23) neurons. ****p < 0.0001, two-sided t test. All error bars in this figure indicate s.d. **f** Fiber photometry recording setup. GCaMP6s was expressed in SOM neurons. **g** Average calcium signals in response to sucrose (red) or water only (black) (2 animals, 25 trials in each animal). Shades indicate s.e.m. **h** Top, slice recording from tdTomato-labeled MS SOM neurons while photo-stimulating axon terminals from ChR2-expressing PBN neurons. Bottom, LED-evoked EPSC recorded under -70mV holding potential in an example MS SOM neuron, in the presence of TTX and 4AP. Scale bar, 10 pA, 25 ms. **i** Average EPSC amplitudes of 8 out of 19 MS SOM neurons evoked by LED stimulation of PBN axons. Error bars represent s.d. **j** Optogenetic activation of SOM or PV neurons in MS in behaving mice. **k** Movement tracks for an example animal in RTPP test with photoactivation of SOM neurons. **l** Percentage time spent in the LED-On chamber for GFP control and ChR2-expressing groups. ***p = 0.0002, two-sided t test, n = 10 mice in each group. **m**, **n** RTPP test with photoactivation of PV neurons. **p = 0.001, two-sided t test, n = 10 mice in each group. Source data are provided as a Source Data file.

(Fig. 9a). We further expressed ChR2 in MS SOM neurons and photostimulated the ChR2-expressing axon terminals in LHb (at 20 Hz) bilaterally via implanted optic fibers above LHb (Fig. 9b). The stimulation-induced place preference in the RTPP test (Fig. 9c, d), indicating that the MS SOM to LHb projection is rewarding. In addition, we expressed Cre-dependent halorhodopsin (AAV-DIO-eNpHR3.0) in MS SOM neurons and photoinhibited their axons in LHb bilaterally by delivering yellow light covering the entire CS plus US window over 5 days' conditioning sessions. (Fig. 9e). The inhibition of the MS$_{SOM}$ → LHb projection resulted in significant impairment of reward associative learning (Fig. 9f–h). In addition, the licking behavior during the sucrose consumption window was reduced (Supplementary Fig. 6c). These data demonstrate that

indeed MS SOM neurons mediate the reward associative learning through their projection to LHb.

## Discussion

In this study, using cell-type specific analysis we have discovered a bottom-up ascending sensory pathway (Fig. 9i) mediated by MS SOM neurons to relay rewarding taste signals. The activity of the SOM neurons was increased by rewarding stimuli, and precise optogenetic interventions revealed that the activation of these neurons was causally linked to appetitive/approaching behaviors. Positive reinforcement learning resulted in a plastic change (emergence and enhancement) in their responses to CS. Temporal silencing of the SOM neurons largely impaired the positive reinforcement

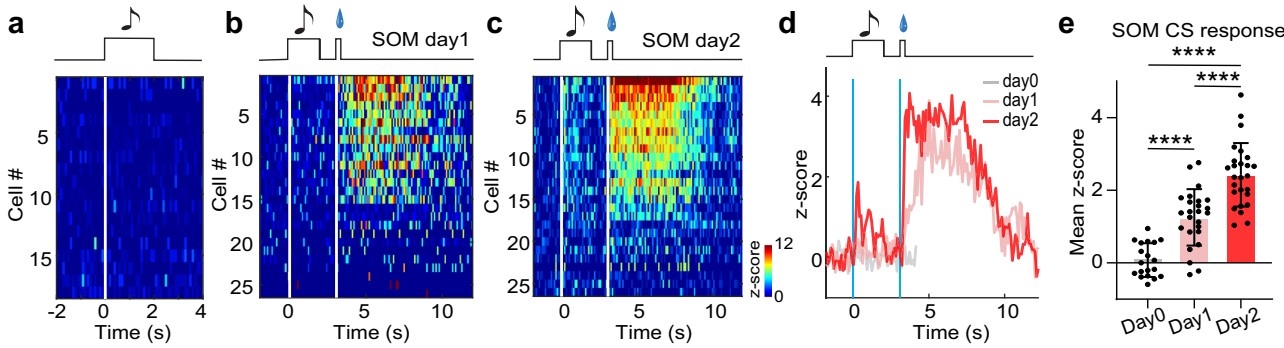

**Fig. 7 SOM neurons increase their responses to the reward-predicative cue during learning. a** Heatmap plot for responses of SOM neurons ($n = 18$) to a tone (16 kHz) at day 0, which was used as CS in the upcoming training sessions. **b**, **c** Responses of SOM neurons during day 1 (**b**) and day 2 (**c**) training sessions ($n = 26$ cells). **d** Population average of Z-score. **e** Mean Z-score for tone responses before training and during day 1 and day 2 training sessions. ****$p < 0.0001$, one-way ANOVA with post-hoc Tukey's test, two-sided. Source data are provided as a Source Data file.

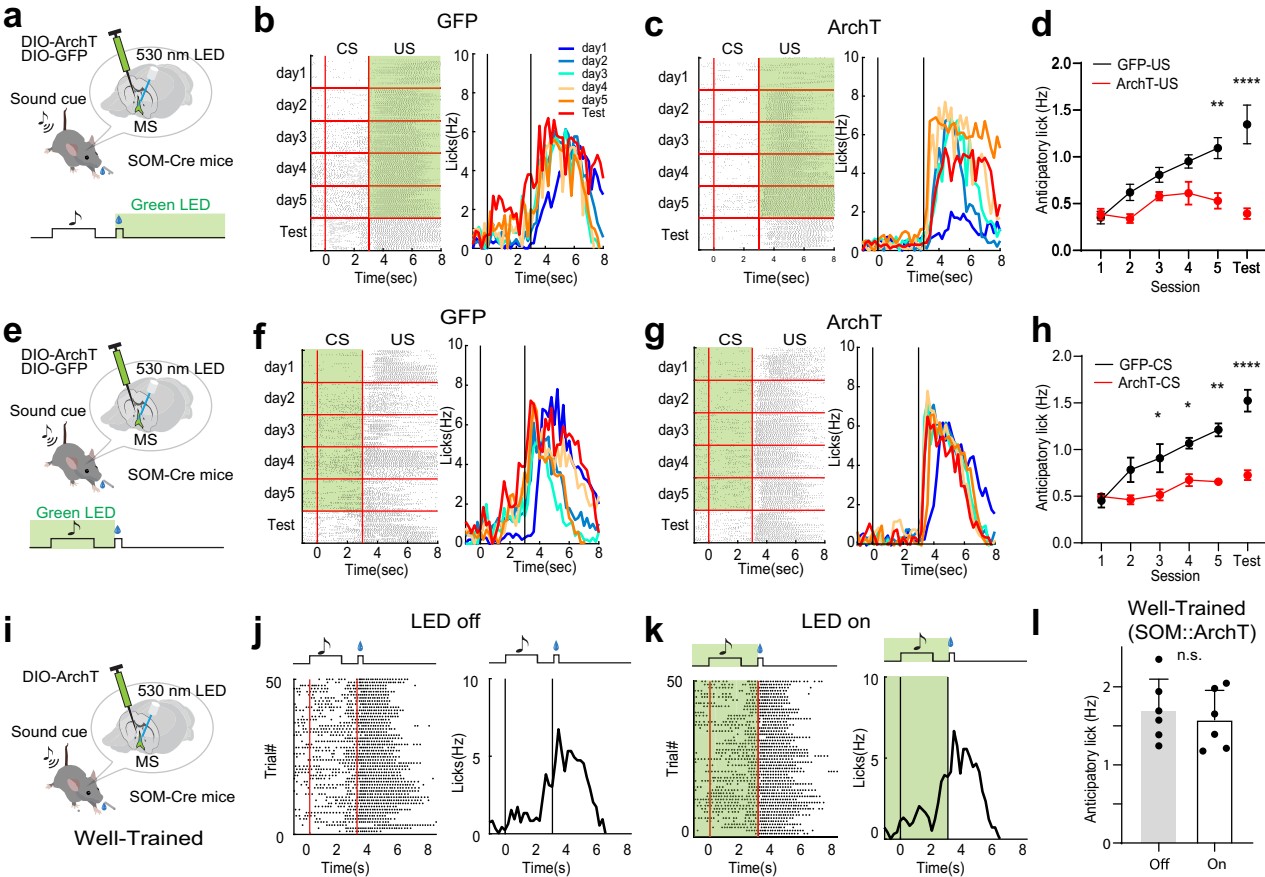

**Fig. 8 SOM neurons are required for reward associative learning. a** Optogenetic inhibition of MS SOM neurons during the US window (green shade). **b** Raster plot of licking events (left) and time-dependent lick rates (right) for different training sessions (day 1-day 5) and the test session after training for an example GFP control mouse. Note that green LED light was not applied on the 6th day (Test). **c** Similar to (**b**), but for an ArchT-expressing mouse. **d** Average anticipatory lick rates for GFF control (black, $n = 3$) and ArchT (red, $n = 3$) mice. D1, $p > 0.9999$, D2, $p = 0.2758$, D3, $p = 0.59$, D4, $p = 0.1058$, D5, **$p = 0.0019$, Test, ****$p < 0.0001$, two-way ANOVA with post-hoc Bonferroni test, two-sided. All error bars in this figure indicate s.e.m. **e** Optogenetic inhibition of SOM neurons during the CS window (from 1 s preceding the CS onset till sucrose onset). **f** Lick rates for a GFP control animal. **g** Lick rates for an ArchT-expressing animal. **h** Average anticipatory lick rates for GFP control (black, $n = 3$) and ArchT (red, $n = 3$) mice. D1, $p > 0.9999$, D2, $p = 0.1214$, D3, *$p = 0.0319$, D4, *$p = 0.0306$, D5, **$p = 0.0011$, Test, ****$p < 0.0001$, two-way ANOVA with post-hoc Bonferroni test, two-sided. **i** Optogenetic inhibition in well-trained animals. **j** Raster plot of licking events (left) and time-dependent lick rates (right) in LED-off trials for an example ArchT mouse. **k** Similar to (**j**), but for LED-on (green shade) trials. **l** Average anticipatory lick rates for LED-off and LED-on trials ($p = 0.318$, two-sided paired $t$ test, $n = 6$ mice). Source data are provided as a Source Data file.

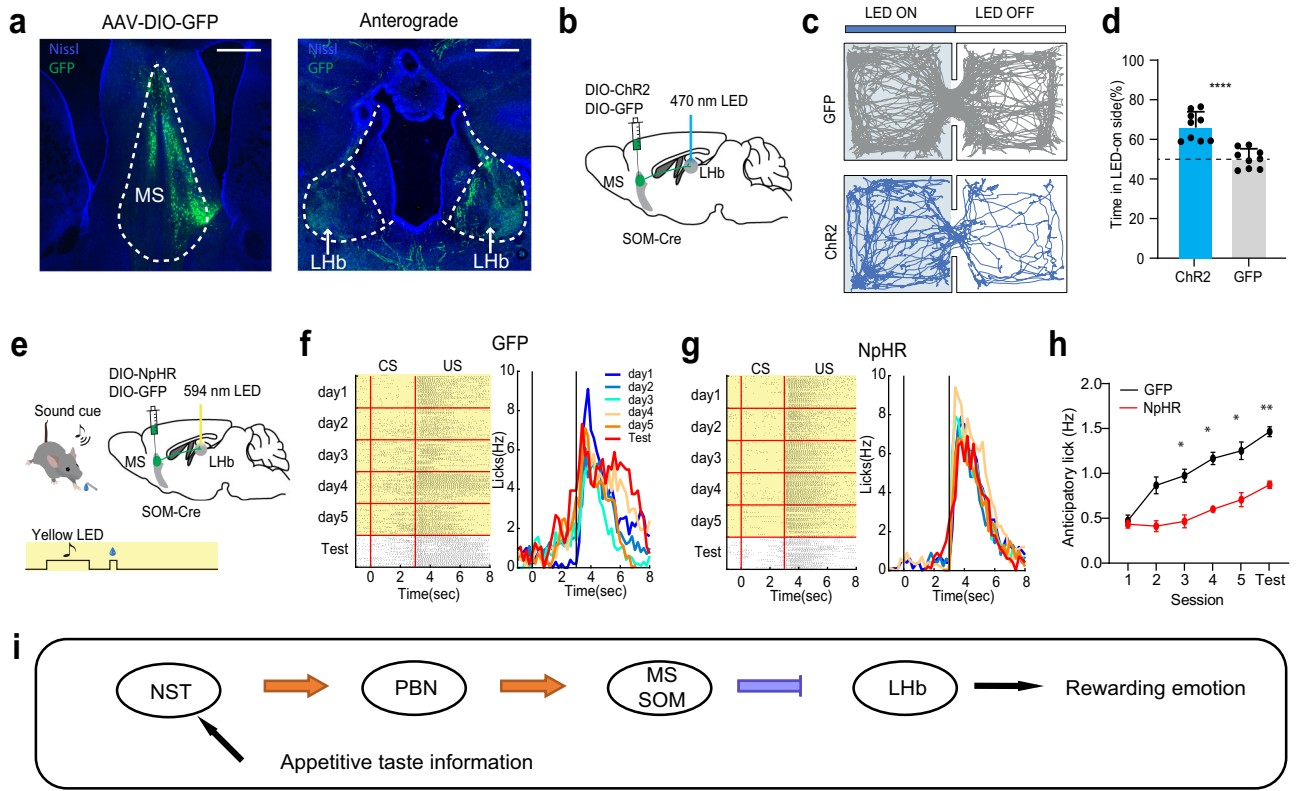

**Fig. 9 The MS SOM-LHb pathway mediates reward associative learning. a** Left, GFP-labeled MS SOM neurons by injecting AAV-DIO-GFP in SOM-Cre mice. Right, GFP-labeled axons in LHb. Scale bar, 500 μm. **b** Photoactivation of MS SOM axons in LHb (bilateral). **c** Movement tracks of a GFP control and ChR2-expessing animal in the RTPP test. **d** Percentage time spent in the LED-on chamber for GFP and ChR2 groups. ****$p < 0.0001$, two-sided $t$ test, $n = 9$ mice for each group. Error bars represent s.d. **e** Optogenetic inhibition of MS SOM axons in LHb (bilateral) during the entire reward associative learning window (yellow shade). **f** Raster plot of licking events (left) and time-dependent lick rates (right) for different training sessions and the test session for an example GFP control mouse. **g** Similar to (**f**), but for an NpHR-expressing mouse. **h** Average anticipatory lick rates for GFP control (black, $n = 4$) and NpHR (red, $n = 4$) mice. D1, $p > 0.9999$, D2, $p = 0.054$, D3, *$p = 0.0147$, D4, *$p = 0.0139$, D5, *$p = 0.0325$, Test, **$p = 0.0012$, two-way ANOVA with post-hoc Bonferroni test, two-sided. Error bars indicate s.e.m. **i** Schematic bottom-up pathway for transmitting appetitive taste information via NST, PBN and MS SOM neurons before reaching LHb, resulting in positive emotion. Source data are provided as a Source Data file.

learning, by intervening either the CS- or US-related activity of these neurons. These results have revealed physiological properties of a cell-type specific, reward-coding pathway that underlies specific appetitive reinforcement learning.

**MS transmits divergent valences and bidirectionally modulates LHb.** MS receives aversive auditory and somatosensory signals from the pontine central gray (PCG) and transmits negative valence information to LHb via its glutamatergic neurons[7,8]. In the present study, we found that MS GABAergic neurons, specifically SOM inhibitory neurons, receive rewarding gustatory signals from PBN, which is also in the pons. Thus, the SOM neurons are selectively activated by bottom-up rewarding sensory stimuli, while the glutamatergic neurons are selectively activated by aversive sensory stimuli. It remains to be investigated whether the SOM neurons can be activated by a broad range of rewarding stimuli, but at least the observation of their activation by water consumption (Fig. 6g) suggests that the rewarding stimuli are not limited to taste modality. Since MS GABAergic and glutamatergic axons both project to LHb and can co-innervate the same LHb neuron[7], MS can convey both rewarding and aversive signals to LHb through these parallel and convergent projections, which then bidirectionally modulates LHb neuronal activity through their opponent interactions. Thus, in addition to previously proposed function roles of MS in speed coding[41], sleep control[19], generation of hippocampal theta rhythms[42], exploratory

behavior[43], and attention[44], we further demonstrate that this structure also serves as an important hub for processing valence of sensory information.

LHb has been shown to play a critical role in mediating aversion/reward[10,11,14] through top-down modulation of dopaminergic and serotonergic systems[17,18], and is strongly implicated in major depressive disorders[9,12]. Accumulating evidence supports that attenuating neuronal activity in LHb can provide therapeutic effects[9]. Because there are few GABAergic neurons within LHb[13], in physiological conditions the suppression of LHb relies on its afferents which are GABAergic. Parallel excitatory and inhibitory projections may be used as a common strategy for input sources to LHb, so that the same or cross modality input signals (internal or external) can bidirectionally modulate LHb neuronal activity depending on the balance between the excitatory and inhibitory presynaptic neurons. As such, MS may serve as an important hub where bottom-up sensory signals with opposite valences converge. Through MS-LHb mediated pathways, external sensory cues of different modalities can then be transformed into emotional valences and affect the mood and behavior of animals.

**A non-canonical taste-related valence processing pathway.** Taste sensation is essential for animal to discriminate and evaluate the palatability of food cues and to approach nutritional food or avoid toxic substances. Once sensed by the peripheral taste

receptor cells, gustatory information is first relayed by taste ganglia and then NST[45]. As a direct downstream target of NST, PBN broadcasts the gustatory information to many regions throughout the brain, including the taste thalamus, gustatory cortex, amygdala, and hypothalamus[29,30]. Sweet and bitter are two basic taste qualities with opposite valences, i.e. appetitive and aversive respectively. However, whether valences of sweet and bitter taste are represented by molecularly different populations of neurons and where this segregation occurs have remained open questions. Sweet and bitter taste qualities are represented by different cortical fields in the gustatory cortex and such segregation remains in the cortical projections to the amygdala[46–48], with the latter considered to be able to drive valence-specific taste behaviors[48]. Nevertheless, it has been proposed that each taste modality specified by a taste receptor type has its own neuronal pathway[45,49,50], implying that segregation of taste quality or valence can occur before the stage of the cortex. Indeed, separate neuronal populations in PBN are reported to encode appetitive and aversive taste stimuli respectively[22,33]. Therefore, the appetitive and aversive taste information may have already been segregated in PBN and then further utilized by its downstream targets to regulate emotional states and related appetitive/aversive behaviors. Our results demonstrate that MS SOM neurons specifically receive appetitive taste information from PBN and encode its valence value. Such information is further relayed to LHb important for reward processing. Therefore, the NST → PBN → MS → LHb pathway constitutes an ascending gustatory pathway devoted to emotional processing (Fig. 9i), and could be independent of the conventional NST → thalamus→cortex pathway required for taste discrimination. How this pathway interacts with the amygdala in driving valence-dependent taste behaviors needs to be further investigated. It should be noted that PBN projects not only to MS, but also to the lateral septum (LS) (Fig. 3e). In addition, NST, MS and LHb also have multiple projection targets and rich collaterals. The specific NST-PBN-MS-LHb pathway we focus on in this study could be one of many pathways contributing to reward processing of taste signals. This is evidenced by results showing that inactivation of each component of the pathway only partially impeded the licking behavior to sucrose delivery (Supplementary Fig. 6).

**MS is a critical hub for associative learning**. MS is part of the basal forebrain, which is one of the earliest structures that show degenerative changes in Alzheimer's disease and has been implicated in its characteristic memory disfunction[51]. Previous studies have demonstrated involvements of MS in learning and memory, but with a focus on the cholinergic system[52]. In Pavlovian conditioning, the unconditioned stimulus has to contain an inherent valence value (positive or negative), which is then associated with the conditioned stimulus during conditioning so that the latter acquires the same valence. Cholinergic neurons in MS however appear to encode salience but not valence of unconditioned stimuli[7,52], suggesting that other cell types[20] such as glutamatergic and GABAergic neurons may be involved in valence coding. Previous studies suggest that MS glutamatergic neurons encode aversive signals in various modalities[7] and that silencing MS impairs sound-cued fear conditioning[8]. Here, we further demonstrate that the SOM type of MS GABAergic neurons specifically encode rewarding signals and positive valence and that their responses to the unconditioned stimulus are required for the reward-cue association. Therefore, via separate molecularly defined neuronal populations, MS can convey both positive and negative valences of unconditioned stimuli required for associative learning. The CS-US association is mediated at least partially by changes of MS neuron responses to the CS (e.g.

an increase of CS responses during reward learning). The acquired valence of the CS could be relayed from MS to LHb, which contributes to the observed changes of CS responses in LHb[14]. In addition, previous studies have demonstrated that neuronal activity in MS can modulate hippocampal spatial representation via the septal-hippocampal projection[53]. Therefore, it is likely that MS also provides valence signals to the hippocampus and contributes to the formation of reward-associated spatial memory[54]. Notably, in the septal-hippocampal pathway, MS GABAergic neurons have been shown to mostly innervate GABAergic interneurons in the hippocampus[38,55]. Thus, activation of MS GABAergic neurons may result in disinhibition of pyramidal neurons in the hippocampus, facilitating the formation of reward site-specific place fields[54]. Together, these results suggest that MS may act as a critical hub for both aversive and reward learning, by receiving sensory inputs of multiple modalities and distributing associated valence values to various downstream structures. This view fills in a gap in our understanding of roles of MS in learned behaviors.

**Plasticity of SOM neurons and their role in reward learning**. The functional roles of SOM neurons in the medial septum complex have been poorly studied. Previous investigations on the function of SOM neurons around the region primarily focused on the population residing mostly in the lateral NDB, which is more ventral and lateral to the area we focused on in this study. The neurons are found to powerfully inhibit all other cell types and gate the basal forebrain input to the cortex[19,56]. A recent study has shown that optogenetic inactivation of basal forebrain SOM neurons impairs spatial working memory[57]. Moreover, SOM neurons in the hippocampus and cortical regions have been shown to undergo anatomical and functional changes in experience-dependent plasticity[58]. Our previous study suggests that MS neurons in general do not respond to tones, while selectively respond to broadband noise with a high intensity threshold[8]. That SOM neurons do not respond to a CS tone in naïve mice (Fig. 7a) is consistent with this previous observation. After conditioning, the SOM neurons acquire responses to the reward-predicative tone, suggesting that the initially subthreshold tone-evoked input to these neurons has been strengthened and become suprathreshold through mechanisms such as long-term potentiation (LTP). Pairing a subthreshold input and a suprathreshold input with an appropriate temporal order can lead to LTP of the subthreshold input[59,60]. That both the CS (subthreshold) and US (suprathreshold) responses of SOM neurons are required for the acquisition of learning is consistent with the idea that LTP-like synaptic plasticity is a cellular substrate for the formation of CS-US association[61,62]. The conditioning-induced enhancement of CS responses has also been observed in the basal forebrain[63,64], as well as in many other brain regions[5,65,66].

The finding that MS SOM neuron activity is required for the formation of reward association memory but not for the retrieval of this memory (Fig. 8i–l) suggests a dissociation of neural circuits for the acquisition and expression phase of reward learning. Possibly, after conditioning, the plasticity of CS responses is transferred to other neuronal substrates, which can then support MS-independent behavioral responses to the CS tone alone. Similar dissociation has been reported elsewhere. For example, the paraventricular thalamic nucleus (PVT) is involved in the acquisition but not expression of associative memory[37], while dopamine neurons in the dorsal raphe only affect the expression phase of associative learning[67].

In the reward processing system, the activity of some neurons such as dopamine neurons is proposed to encode reward prediction errors[65,68]. For this type of neurons, reward-predictive

cues gain the ability while the predicted reward gradually fail to activate them during learning. However, in our results, after learning the predicted reward remains to be able to activate the MS SOM neurons (Fig. 7d), suggesting that activity of the SOM neurons does not serve as reward prediction errors. In addition, the SOM neurons do not maintain a high-level activity during the anticipatory phase (i.e. delay period) of reward responses, unlike serotonergic neurons in the dorsal raphe[69,70]. Through the inhibitory projection to LHb, the plasticity of MS responses to reward-predictive cues can be relayed directly to LHb, which accounts for the enhanced suppressive responses to the cues in LHb[14].

In summary, the results of the current study reveal an important role of MS in associative learning. By broadly receiving multisensory inputs of both rewarding and aversive nature, MS integrates and broadcasts positive and negative valences of sensory cues via different neuronal populations, which are required for the formation of related associative memory.

## Methods

All experimental procedures in this study were in accordance with the guidelines for the care and use of laboratory animals of the US National Institutes of Health (NIH), and were approved by Animal Care and Use Committee (IACUC) of the University of Southern California.

**Animals**. Experiments were performed in adult (2–3 months old) male and female mice. Wild-type (C57BL/6J) and transgenic (Vgat-*ires*-Cre, Vglut2-*ires*-Cre, SOM-*ires*-Cre, and PV-*ires*-Cre) mice were obtained from the Jackson Laboratory and were housed with a 12 h light-dark cycle, at 65–75 °F temperature and 40–60% humidity. All recordings and behavioral tests were conducted in the dark cycle.

**Virus**. AAV1-EF1α-DIO-hChR2(H134R)-EYFP-WPRE (UPenn vector core, Addgene, 20298), AAV1-CAG-FLEX-ArchT-GFP (UNC GTC vector core, Addgene, 29777), AAV1-CAG-FLEX-GFP-WPRE (UPenn vector core, Addgene, 51502), AAV5-EF1a-DIO-hM4D(Gi)-mCherry (Addgene, 50461), AAV5-EF1a-DIO-mCherry (Addgene, 50462), AAV1-CA-FLEX-RG (Addgene, 38043), AAV1-EF1α-FLEX-TVA-mCherry (Addgee, 38044), EnvA-G-deleted Rabies-GFP (Salk vector core), AAV9-Syn-FLEX-GCaMP6s-WPRE-SV4 (Addgene, 100845), AAV1-hSyn-hChR2(H134R)-EYFP (Addgene, 26973), and AAV1-EF1α-DIO-eNpHR3.0-EYFP (Addgene, 26966) were used in this study.

**Surgical procedures**. Stereotaxic injections of virus were carried out as we previously described[7,8]. The mouse was anesthetized with isoflurane (1.5–2% by volume). A heating pad was placed underneath the animal body to maintain the body temperature of the animal. A small incision was made on along the midline to expose the skull. One ~0.2 × 0.2 mm² craniotomy window was made for the target region (MS: 0.98 mm anterior to the bregma, 1 mm lateral to the midline, 4.25 mm below the pia with a 13.5° angle; PBN: 5.2 mm posterior to the bregma, 1.25 mm lateral to the midline, 2.8 mm below the pia with 0° angle; NST: 7.5 mm posterior than the bregma, 0.3 mm lateral to the midline, 3 mm below the pia with 0° angle; LHb: 1.5 mm posterior than the bregma, 1 mm lateral to the midline, 2.5 mm below the pia with a 10° angle). The adeno-associated viruses (AAVs, encoding ChR2, GFP, ArchT, NpHR, or GCamp6s) or pseudo-typed rabies virus (encoding GFP) were used depending on the purpose of experiments and strain of mice. Virus was delivered through a pulled glass micropipette (inner diameter of tip: ~20–30 μm) using pressure injection via a micropump (World Precision Instruments). For each injection, 60 nL of viral solution was injected at a rate of 15 nL/min. Right after the injection, the pipette stayed for 4 min before withdrawal. The scalp was then suture closed and the animal was administered ketoprofen (5 mg/kg) and buprenorphine (0.5 mg/kg) to minimize inflammation and discomfort. Animals were recovered from anesthesia on a heating pad and then returned to their home cages.

**Optogenetic preparation and stimulation**. 2 weeks after the virus injection, an optic cannula (200 μm core, RWD Inc.) was stereotaxically implanted above the MS or LHb. After at least 1 week of recovery, animals were habituated to connecting to an optic fiber cord. For photoactivation, 20 Hz (5 ms pulse duration) light stimulation was delivered through the optic fiber cord which was connected to a blue LED source (470 nm, Thorlabs). For photoinhibition, a sustained green LED light (530 nm, Thorlabs, for ArchT) or yellow light (594 nm, Thorlabs, for NpHR) was applied during the entire period for inhibition. The efficiency of photoactivation and photoinhibition has been verified in slice recording and in vivo optrode recording experiments. The LED power measured at the tip of the fiber (connected with the optic cannula) is around 3–5 mW.

## Behavioral tests

*Spatial reward learning*. Spatial reward learning was performed with a hole-board, where sucrose water was filled in a target hole. Behavioral tests were performed after water deprivation for 24 h. The animal was placed on a platform which was connected with one corner of the hole-board. Time counting was started when the animal left the platform and entered the board, and ended when it found the target hole and started to lick the water. 5 trials were given each day for 4 consecutive days. Compensatory water was given to the animal after daily trial so that the animal maintained > 90% of the original body weight.

*Real-time place preference*. Real-time place preference test was performed as described before[7]. A clear acrylic behavior box (40 cm × 20 cm × 20 cm, divided into two chambers, put in a larger white foam box) with normal bedding materials was used. For each trial, the mouse was initially placed in the non-stimulation chamber, and LED (480 nm, 10 Hz, 5-ms pulse duration) stimulation was constantly delivered once the animal entered the stimulation chamber and was stopped once the animal exited. The total duration of each test session was 20 min. Animals were returned to their home cage after each test session. The stimulation chamber was randomly assigned and balanced for the whole group of animals. This test was controlled by a customized close-loop optogenetic control system with online real-time mouse detection software and a computer-controlled Arduino microcontroller (https://www.arduino.cc/) described in detail below.

*Self-stimulation*. Mice were placed in an operant box equipped with two ports for nose poke at symmetrical locations on one of the cage walls. The ports were connected to a photo-beam detection device allowing for measurements of responses. A valid nose poke at the LED-on port lasting for at least 500 ms triggered a 1 s long 20 Hz (5-ms pulse duration) LED pulse train delivery controlled by an Arduino microcontroller. The LED-on port was randomly assigned and balanced within the group of tested animals. The test lasted for 40 mins. Video and time stamps associated with nose poke and laser events were saved in a computer file for post-hoc analysis.

*Light-dark box test*. An acrylic behavior box (40 cm × 20 cm × 20 cm) was divided into a dark chamber (10 cm × 20 cm × 20 cm) and a light chamber (30 cm × 20 cm × 20 cm). The dark chamber was shielded with black aluminum foil. A small opening located at floor level in the center of the dividing wall allowed the animal to freely move between the dark and light chambers. To prevent hindering movements of the optic fiber cable, a narrow opening was made on the divider between the two chambers. Besides that, on top of the light side chamber, two curved metal wires were placed to guide the movement of the optic fiber cable. Animal was placed in the light side at the beginning and its behavior was recorded by a camera above the box. The time spent on the light side was analyzed using the object-detection software described below.

**Real-time animal detection and closed-loop optogenetic control**. A customized mouse detection software was used for online real-time animal detection (written by Guang-Wei Zhang, in Python 3.4, www.python.org, with OpenCV library, https://opencv.org). The behavior of the animal was monitored using an infrared camera at 24 frames per second. Each video frame was gaussian blurred and then binarized. The centroid of the detected contour was used to determine the location of the animal. In the two-chamber place preference test, once the mouse entered the assigned LED-on chamber, computer-controlled Arduino microcontroller (www.arduino.cc) generated TTL signals to drive the LED light source (ThorLabs Inc.). The behavior test was run automatically without the experimenter's interference and the result was quantified right after each experiment.

**Intraoral infusion of sucrose/quinine and Pavlovian conditioning**. For passive sucrose or quinine delivery, a soft silastic tubing was subcutaneously inserted to the oral cavity of the mouse through a small incision on cheek[69]. The tube was adhesive to the cheek with sutures. This approach allowed us to precisely control the time and volume of sucrose or quinine delivered into the mouth. After three days of recovery, a micro pump (Lee-Company) was used to infuse either quinine (5 mM, 10 μL) or sucrose (5% w/v) into the oral cavity through the tube.

For active licking, mice were water-deprived and habituated to the head-fixation condition to lick the waterspout for 2–3 days. Once they reliably licked free water, they were subjected to Pavlovian conditioning. During conditioning, a 16 kHz tone cue was presented for 2 s, followed by a 1 s delay period, and then 10 μL 5% w/v sucrose water was delivered. Each training session contained 50 trials. The inter-trial interval varied between 30–40 s. For each training session, the animal could drink about 1 mL water before returned to home cage, where no water was available. An infrared camera was used to record videos for detection of licking events. An infrared LED was also captured to serve as a trigger signal for each trial and was used to align extracellular recording with video recording. Licking events were detected by a customized software (written by Li Shen, in Python with OpenCV library) and further confirmed by visual inspection. The data were further processed with customized Matlab scripts (Mathworks). The anticipatory lick rate was quantified between the onset of the tone cue and that of sucrose delivery, with the baseline lick rate (before the onset of CS tone) subtracted. The anticipatory lick

rate reached a plateau after 5–6 training sessions, and animals were considered as well-trained. To test the effect of photoinhibition in well-trained animals, optogenetic stimulation was applied in randomly selected 50 trials out of 100 total trials.

**Optrode recording and spike sorting.** Three days before recording, the mouse was anesthetized with isoflurane (1.5–2% by volume). A head post for fixation and another for angle marker were mounted on top of the skull with dental cement and a craniotomy was performed over the MS. Silicone adhesive (Kwik-Cast Sealant, WPI Inc) was applied to cover the craniotomy window until the recording experiment. For pharmacological silencing with muscimol, a drug delivery cannula (internal diameter: 140 μm) was stereotaxically implanted into the target area. After baseline recording, fluorescent muscimol-bodipy (0.7 mM in ACSF with 5% DMSO) was infused via the implanted cannula. Recording session to evaluate silencing effects was started 10 min after the infusion.

Recording was carried out with an optrode (A16-Poly2-5mm-50 s-177-OA16LP, 16 contacts separated by 50 mm, the distance between the tip of the optic fiber and the probes is 200 μm, NA 0.22, Neuronexus Technologies) connected to a laser source (473 nm) via an optic fiber. The optrode was lowered into the MS region and signals were acquired using the Plexon recording system. The Vglut2, Vgat, PV or SOM neurons were optogenetically tagged by injecting AAV-floxed-ChR2 in a corresponding Cre-driver mouse or by crossing a Cre-driver line with a ChR2 reporter line (Vglut2-Cre × Ai27, Vgat-Cre × Ai27, PV-Cre × Ai27, or SOM-Cre × Ai27). To identify ChR2+ neurons, 16-Hz or 32-Hz (5-ms pulse duration) laser pulse trains were delivered intermittently. Extracellular signals were recorded at 30 kHz sample rate and were filtered through a bandpass filter (0.3–10 kHz). The nearby four channels of the probe were grouped as tetrodes, and semiautomatic spike sorting was performed by using Offline Sorter (Plexon). Semiautomated clustering was carried out on the basis of the first three principal components of the spike waveform or Peak-Valley values on each tetrode channel using a T-Dist E-M scan algorithm (scan over a range of 10–30 degree of freedom) and then evaluated with sort quality metrics. Clusters with isolation distance <20 and L-Ratio > 0.1 were discarded[7]. Spike clusters were classified as single units only if the waveform SNR (Signal Noise Ratio) exceeded 4 (12 dB) and the inter-spike intervals exceeded 1.2 ms for >99.5% of the spikes. MS ChR2+ neurons were optogenetically identified by their time-locked spike responses to blue laser pulses. Only spikes with onset latencies (relative to the light pulse) < 4 ms were considered as being directly evoked. The correlation coefficient between the average waveforms of these laser-evoked spikes and sucrose- or quinine-evoked spikes was calculated and only units with cc >= 0.97 were considered as a valid unit.

**Response quantification.** After spike sorting, spike trains for each cell were analyzed with customized MATLAB scripts (Mathworks). Peri-stimulus spike time histograms (PSTHs) with 10 ms time bins were generated. Firing rates were normalized to the baseline activity by calculating a Z-score ($z = (x-\mu)/\sigma$), with μ being the average spontaneous firing rate and σ being the standard deviation in the 2 s window preceding the stimulus onset. For each cell, the evoked response within 2 s after the stimulus onset was compared with the spontaneous activity to determine whether the stimulus could excite or suppress the neuronal activity (unpaired $t$ test, $p < 0.05$). Recording location and virus expression were verified post-hoc with standard histological procedures.

**Slice whole-cell recording.** To confirm the connectivity between PBN glutamatergic axons and MS neurons, Vglut2 -Cre mice injected with AAV1-EF1α-DIO-hChR2(H134R)-EYFP-WPRE in PBN were used for slice recording. To confirm the connectivity between PBN axons and MS SOM neurons, SOM-Cre-Ai14 mice injected with AAV1-hSyn-hChR2(H134R)-EYFP in PBN were used for slice recording. Four weeks following the injection, animals were decapitated following anesthesia and the brain was rapidly removed and immersed in an ice-cold dissection buffer (composition: 60 mM NaCl, 3 mM KCl, 1.25 mM NaH$_2$PO$_4$, 25 mM NaHCO$_3$, 115 mM sucrose, 10 mM glucose, 7 mM MgCl$_2$, 0.5 mM CaCl$_2$; saturated with 95% O$_2$ and 5% CO$_2$; pH = 7.4). Coronal slices at 350 μm thickness were sectioned by a vibrating microtome (Leica VT1000s), and recovered for 30 min in a submersion chamber filled with warmed (35 °C) ACSF (composition:119 mM NaCl, 26.2 mM NaHCO$_3$, 11 mM glucose, 2.5 mM KCl, 2 mM CaCl$_2$, 2 mM MgCl$_2$, and 1.2 NaH$_2$PO$_4$, 2 mM Sodium Pyruvate, 0.5 mM VC). MS neurons labeled with tdTomato (SOM-Cre-tdTomato mice) or surrounded by EYFP$^+$ fibers were visualized under a fluorescence microscope (Olympus BX51 WI). Patch pipettes (~4–5 MΩ resistance) filled with a cesium-based internal solution (composition: 125 mM cesium gluconate, 5 mM TEA-Cl, 2 mM NaCl, 2 mM CsCl, 10 mM HEPES, 10 mM EGTA, 4 mM ATP, 0.3 mM GTP, and 10 mM phosphocreatine; pH = 7.25; 290 mOsm) were used for whole-cell recordings. Signals were recorded with an Axopatch 700B amplifier (Molecular Devices) under voltage clamp mode at a holding voltage of –70 mV for excitatory currents, filtered at 2 kHz and sampled at 10 kHz. Tetrodotoxin (TTX, 1 μM) and 4-aminopyridine (4-AP, 1 mM) were added to the external solution for recording monosynaptic responses to blue light stimulation (5 ms pulse, 3 mW power, 10–30 trials).

For testing the efficacy of ChR2 or ArchT, brain slices were prepared similarly, and whole-cell current-clamp recordings were made from neurons expressing ChR2 or ArchT. A train of blue light pulses at different frequencies (5-20 Hz, 5-ms pulse duration) was applied to measure spike responses of ChR2-expressing neurons. Green light stimulation (5-s duration) was applied to measure hyperpolarizations in ArchT-expressing neurons. In vitro slice recording data were analyzed by using pClamp (Molecular Devices) and customized Python codes.

**Cell-type-specific retrograde tracing of monosynaptic inputs.** To trace monosynaptic inputs to Vgat neurons in MS, AAV1-CA-FLEX-RG and AAV1-EF1α-FLEX-TVA-mCherry were stereotactically injected into MS of Vgat-Cre mice. After two weeks, EnvA-G-deleted Rabies-GFP was injected into MS. The animal was sacrificed one week later. Brain tissue was fixed, sectioned, and imaged using a confocal microscope.

**Fiber photometry recording.** SOM-Cre mice were injected with Cre-dependent GCaMP6s (AAV1-Syn-FLEX-GCamp6s-WPRE-SV4) into the MS (60 nL). An optic fiber (400 μm, NA = 0.5, Thorlabs) was implanted over the MS injection site and secured with dental cement. Following 3 weeks of viral expression, GCaMP6s fluorescence was detected through the optic fiber using a fiber photometry system as described previously[71]. LED light (480 nm, Thorlabs) was bandpass filtered (ET470/24 M, Chroma), focused by an objective lens (Olympus) and coupled to an optical fiber (O.D. = 400 μm, NA = 0.48, 1 m long, Doric), which connected to the implanted optic fiber using a ceramic sleeve. The LED power was adjusted to 0.02 mW at the tip of the optical fiber. The fluorescent calcium signal was bandpass filtered (ET525/36 M, Chroma) and collected by a photomultiplier tube (H11706-40, Hamamatsu). Before digitized (250 Hz sampling rate) by a data acquisition card (PCI-MIO-16E-4, National Instruments), the signal was amplified (Model SR570, Stanford Research System) and low-pass filtered (30 Hz). Data were obtained and analyzed using custom LabVIEW and MATLAB software.

**Image acquisition.** To check the expression of EYFP, GFP, or mCherry, or electrode tracks (coated with DiI), the animals were deeply anesthetized using urethane (25%) and transcardially perfused with phosphate-buffered saline (PBS) and paraformaldehyde (4% in PBS). Coronal brain sections (150 μm) were made with a vibratome (Leica Microsystems) and stained with Nissl reagent (Deep red, Invitrogen) for 2 h at room temperature. Each slice was imaged under a confocal microscope (Olympus).

**Statistics.** Pilot experiments were conducted to determine the sample size. Mann-Whitney test was used for non-normality data. Kolmogorov-Smirnov test was used for comparing the difference between optogenetic or chemogenetic silencing and control group. One-way ANOVA and Two-way ANOVA with post-hoc Tukey's test or Bonferroni test were used to test significance between samples. For two-group comparison of normality data, significance was determined by using $t$ test. Paired $t$ test was used to compare data from the same neuron or the same animal.

**Reporting summary.** Further information on research design is available in the Nature Research Reporting Summary linked to this article.

## Data availability

The full datasets generated in the current study are available from the corresponding author upon requests. Source data are provided with this paper.

## Code availability

Animal detection and lick detection programs are available on https://github.com/li-shen-amy/behavior. DIO for the repository is https://doi.org/10.5281/zenodo.5992295.

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

## Acknowledgements

This work was supported by grants from the U.S. National Institute of Health (EY019049 and MH116990 to H.W.T.; DC008983 and MH116990 to L.I.Z.).

## Author contributions

H.W.T., L.I.Z., and L.S. designed the experiments. L.S., G.W.Z., and M.B.S. performed behavioral experiments. L.S. and G.W.Z. performed in vivo electrophysiology and photometry experiments. L.S. and G.W.Z. performed data analysis. C.T. performed slice recording experiments. J.J.H. and N.K.Z. helped with viral injections and in vivo experiments. H.W.T. and L.I.Z. supervised the project. H.W.T., L.S., G.W.Z., and L.I.Z. wrote the manuscript.

## Competing interests

The authors declare no competing interests.
