## [Peer Review File · Nature Communications]

REVIEWER COMMENTS

Reviewer #2 (Remarks to the Author):

The evidence to support the conclusions is weak in this study.

There are some technical concerns, and important controls are missing, which have been stated in the comments to authors.

Reviewer #3 (Remarks to the Author):

This is an extensive and potentially important work demonstrating the relationship between the medial septum (MS)-associated circuitry and reward motivated behavior. The study involves circuit tracing and manipulation techniques, different behavioral approaches, and measurements of MS activity using electrophysiological recordings and fiber photometry. It is proposed that a circuit involving excitatory projections from NTS to PBN and PBN to MS, and an inhibitory SOM projection from MS to lateral habenula, relays “intrinsically rewarding taste signals”. The data are well-presented but I am not yet convinced in the interpretation. Clarifying these issues would strengthen the paper:

1. There are many behavioral tests and activity measurements that are interspersed within a single figure. It would be much clearer if neuronal activity and behavioral data collected from the same paradigms were grouped together.
2. Fig 1: a-g convincingly shows that optogenetic stimulation of MS induces behavior consistent with reward. H-o show elevated MS activity in response to sucrose and reward. This is followed by a demonstration, using vGlut -and vGAT-Cre mouse lines, that such activity is mediated by MS interneurons. This seems clear and straightforward, however, in the later figures, optogenetic silencing only affects pre-sucrose licking but not licking during sucrose exposure (Fig. 3,6,7). How is this explained? If innate reward is mediated by this system one would expect effects of MS silencing on sucrose licking behavior independently of the tone-sucrose associations.
3. Fig. 2 a-p shows that inactivation of PBN and NST decreases MS neuronal responses to sucrose, whereas inactivation of MS enhances lat habenula responses to sucrose. There is a missed opportunity here to link these activity changes to behavior (sucrose licking).
4. Fig. 3 See point 2.
5. Fig. 4 – electrophysiological responses in MS are similarly elevated both during tone and sucrose. In the fiber photometry experiments responses to sucrose are much stronger than responses to tone. How is this discrepancy explained and what does it mean?
6. Fig 5: same issue as point 5.
7. Fig. 6: If SOM is not required for expression of learned responses (just for encoding), how is it related for innate responses to sucrose if at all?
8. Fig 7: same as before: is MS input to lat habenula involved only in reward associations or also in other aspects of reward.

In summary, the most confusing parts of the study relate to the distinction of MS SOM neuron

involvement in innate vs learned (associated) taste reward and in the differences in data on MS neuronal activity collected with different methods. Also, this reviewer subscribes to the view that stimuli are not intrinsically rewarding (see work of Kent Berridge) but only when assigned positive value during processing in the brain. This is especially true when studies are carried out under water deprivation conditions, as it was done here.

** See Nature Research's author and referees' website at www.nature.com/authors for information about policies, services and author benefits.

Responses to reviewers:

We thank the reviewers for their time and effort in reviewing our manuscript “A Bottom-up Reward Pathway Mediated by Somatostatin Neurons in the Medial Septum Complex Underlying Appetitive Learning”. We appreciate the encouraging and constructive comments on our work. We have made every effort to address the concerns raised by the reviewers. Our point-by-point response (in blue) is provided below each of reviewers’ comments. The major changes in the text are labeled in red color.

Responses to Reviewer #1

This manuscript by Shen et al. examines novel roles in sucrose reward for the poorly understood somatostatin/GABA neurons in the medial septum (MS). They show that MS SS neurons are activated by sucrose and (after learning) by cues that predict sucrose. PBN afferents to the MS play crucial roles in these signaling patterns, while MS SS efferents to the LHb drive inhibitory responses to sucrose in that target. Overall, the PBN-MS-LHb circuit is responsible for some learned behavioral responses to sucrose such as approach and anticipatory licking.

The findings are elegant, and the experiments logical and well-executed, and employ an impressive combination of advanced techniques. The results/discussion well-written, and methods described in detail. My only major concerns are the relatively weak documentation of anatomical placement sites. Because SS neurons are present in both the MS and immediately adjacent lateral septum (LS), and in fact are even more abundant in the LS, the authors should clearly document that they have distinguished the LS from the MS in placements of viral infections and optical/electrophysiological probes. As written, distribution of virally infected SS neurons is shown in only one experiment (Fig. 7), and placements of probes are not documented at all.

We have clarified in the Methods that the viral expression as well as placements of optic fibers and recording electrodes have been verified for each experiment with post hoc histology/imaging. In the revised manuscript, we have provided such documentations in **Extended Data Figure 1**.

A related anatomical issue is that the discussion in some areas (e.g. line 343) refers to studies of different subsets of SS neurons than the MS without clearly stating as such. Line 343 cites a study that examines SS neurons in the “basal forebrain”, but the cited work shows these neurons to be a different population residing more ventrally and laterally than the MS. Readers not already familiar with the cited work might be led to believe this work studies an overlapping region, rather than a distinct population.

This is a good point. In the revised manuscript, we have clarified that the area infected in our study included the medial septum and the medial diagonal band nucleus (NDB) (page 5). The SOM neurons studied in the previous work are located in the ventral NDB and are more ventral and lateral than the SOM cells we studied here (Page 19).

Responses to Reviewer #2:

The study by Shen et al. sought to decipher circuit pathways for appetitive learning. Generally, this is an interesting topic. These experiments were designed well, the data were interpreted well, and the manuscript was written clearly. There are some specific comments as below.

Comments on Figure#1

1a). A series of sections of viral transfections and optic fiber track should be provided to show accurate infections and fiber target to the MS, which is also for other figures.

In the revised manuscript, we have provided additional post hoc histology/imaging data to demonstrate the locations of viral expression, optic fiber and electrode placements for all experiments on VGAT and SOM neurons (**Extended Data Figure 1**).

1b). To exclude the PS-induced non-specific effects, mice transduced with control vectors should be performed as an important control.

We thank the reviewer for this helpful suggestion. In the revised manuscript, we have provided additional experimental data from control mice injected with AAV-DIO-GFP (**Figure 1b&1c**). There was no bias towards either port in the GFP control mice, arguing against any non-specific effects generated by photo-stimulation itself.

1d) Please justify how the photo stimulation was performed in the LDB test. It is not feasible to perform photostimulation on mice with a tethered cable during animal explorations through a small opening on the bottom between the light chamber and dark chamber of the LDB.

We have now provided more details of the experiment in the Method section, “To prevent hindering of movements of the optic fiber cable, a narrow opening was made on the divider between these two chambers. Besides that, two curved metal wires were placed on top of the light-side chamber to guide the movement of the optic fiber cable.” (page 35). Meanwhile, we have added a schematic graph in **Figure 1d** to illustrate these details.

1h-o) The Optrode approach was largely applied for most of the experiments in this study. It was not clearly described regarding how to relate the neuron populations with PS-evoked spikes to those sensitive to sucrose. I believe there would be a mismatch between the populations. Fiber photometry experiments are needed to define which populations including GABAergic and glutamatergic neurons would respond to sucrose or other stimuli. This concern is also for other figures.

We have provided additional information to address this issue. In the revised Figure 2, we have compared response properties between the ChR2-tagged and untagged populations. The PS-responsive (ChR2-tagged) group would all be Vgat+, while the unresponsive group (untagged) would contain largely Vgat- cells, although some uninfected Vgat+ cells may also be included. In the first group, 79% of cells were activated by sucrose, while in the second group only 24% were activated by sucrose (**Figure 2c**). In all units that were activated by sucrose, 72% were tagged and 28% were untagged (revised **Figure 2d**). As reasoned above, the mismatch between tagged and sucrose-activated populations can be attributed to the fact that the untagged population could also contain some uninfected GABAergic neurons since the AAV viral transduction efficiency would not be 100%. Nevertheless, these numbers further support the point that the Vgat+ population preferentially responds to sucrose, while Vglut2+ neurons in general don't respond to sucrose (**Figure 2g**).

The optrode recording approach in general can generate more information than the suggested photometry experiment in that response properties are elucidated at the individual-cell level, besides an overall understanding of the population. This is particularly important for studying a functionally heterogeneous neuronal population. The potential heterogeneity within the same molecularly defined group, such as revealed here in the ChR2-tagged GABAergic population, cannot be revealed by photometry.

Comments on Figure#2

2a, c, g, k, o) Images of viral transfection and /or fiber or electrode track are needed.

We have provided these images in **Extended Data Figure 1**.

2e) It looked that the LS, MS and VDB were retrograde traced, not only the MS. Why did the authors not consider potential roles played by the LS and VDB? A series of sections of the images would demonstrate which regions were involved.

As shown in **Extended Data Fig.1a**, our viral expression mainly affected the medial septum and the medial part of diagonal band nucleus (NDB). We agree that as PBN also projects to LS and ventral NDB (**Figure 3e**) we could not exclude the possibility that the PBN projections to these areas also contribute to reward processing. In the revised manuscript, we have acknowledged this possibility (page 17).

2f) Please clarify the long decay duration for the EPSC, which was about or over 200 ms. CNQX would help define whether it was AMPA receptor-mediated or not.

In the slice recording experiments, we observed light-evoked EPSCs with both short and relatively longer decays (see two examples in **Extended Data Figure 2**). Both could be completely blocked by the application of CNQX (**Extended Data Figure 2**). These data confirmed that the inward synaptic currents were mediated by AMPA-receptors.

2g-r) What were the intervals between before and after the muscimol treatment? Was it possible for the same animals to habituate to the repeated sucrose treatments? To test it, the same experiments are needed to repeat on vehicle-treated mice as controls.

This is a good point. We have clarified in the revised manuscript that the interval was 15 min before and after applying muscimol (page 8). We have performed vehicle-control experiments as suggested. As shown in **Extended Data Figure 3**, licking behavior was not changed after vehicle application. Therefore, there was no obvious habituation. The mouse was water deprived and the consumption volume during the test in fact did not satisfy the animal's need, which is the reason why we needed to provide additional water after the test (page 35).

Comments on Figure#3

Electrophysiology and/or photometry experiments are needed to verify CNO inhibition of hM4Di-transduced neurons.

In **Extended Data Figure 5**, we have provided *in vitro* electrophysiology evidence that application of CNO could reliably suppress spiking activity of hM4Di-expressing MS neurons.

Comments on Figure#4

4a) Please show the track of the inserted electrode to demonstrate the recording location. As stated above, please also clarify the spikes were recorded from GABAergic neurons.

In **Extended Data Figure 1**, we have included images showing the tracks of electrodes. We have now clarified both in the figure and legend (now **Figure 5b-e**) that the recorded spikes were from GABAergic neurons.

Comments on Figure#5

5a-e) Photometry experiments would provide convincing evidence.

Similar to what we have discussed earlier in response to comments on Figure 2, the optrode recording experiment can generate more information than the suggested photometry experiment. Our data show that the majority of optogenetically identified SOM neurons (86.4%) exhibited significant evoked responses to sucrose, whereas none of identified PV neurons show an excitatory response to sucrose (now **Figure 6b-e**). However, that there are inhibitory responses in about 13% of PV neurons is unlikely to be revealed by photometry experiments.

5f-s) The PV neurons did not respond to sucrose, as shown in the 5a-e. Did the authors perform the same experiments on PV neurons as those on SOM neurons? which are important to know to draw the conclusions in this study.

As shown in the **Figure 6m-n**, activation of the PV neurons generated aversion in the place preference test, opposite to that of the SOM population, suggesting that PV neurons are involved in encoding negative valence. Together with the result that the PV neurons did not respond to sucrose, these pieces of evidence indicate that the PV neurons are unlikely involved in reward related behaviors and reward learning. We have further clarified this point in the revised manuscript (page 12).

Additional comments:

This study includes the NST, PBN, MS, and LHb. For each brain region, there are many different projection targets and collateral projections, which need to be carefully considered before drawing the conclusions which would need additional evidence to support.

We agree with the reviewer that the NST, PBN, MS and LHb all have various projection targets and rich collaterals. They have been shown to be involved in many different aspects of brain function. In this study, we focus on examining the contribution of the specific NST-PBN-MS-LHb pathway to reward processing, which has not been studied before. In the revised manuscript, we are cautious about the conclusion made and have acknowledged that other possible circuits may also be involved in reward processing and related learning behaviors (page 17).

Responses to Reviewer #3:

This is an extensive and potentially important work demonstrating the relationship between the medial septum (MS)-associated circuitry and reward motivated behavior. The study involves circuit tracing and manipulation techniques, different behavioral approaches, and measurements of MS activity using electrophysiological recordings and fiber photometry. It is proposed that a circuit involving excitatory projections from NTS to PBN and PBN to MS, and an inhibitory SOM projection from MS to lateral habenula, relays “intrinsically rewarding taste signals”. The data are well-presented but I am not yet convinced in the interpretation. Clarifying these issues would strengthen the paper:

1. There are many behavioral tests and activity measurements that are interspersed within a single figure. It would be much clearer if neuronal activity and behavioral data collected from the same paradigms were grouped together.

We agree with the reviewer and now have reorganized the figures as suggested. Specifically, we have separated the behavioral experiments from the electrophysiological experiments in the original Figure 1. We have also added more labels in the figures to make them better organized.

2. Fig 1: a-g convincingly shows that optogenetic stimulation of MS induces behavior consistent with reward. H-o show elevated MS activity in response to sucrose and reward. This is followed by a demonstration, using vGlut -and vGAT-Cre mouse lines, that such activity is mediated by MS interneurons. This seems clear and straightforward, however, in the later figures, optogenetic silencing only affects pre-sucrose licking but not licking during sucrose exposure (Fig. 3,6,7). How is this explained? If innate reward is mediated by this system one would expect effects of MS silencing on sucrose licking behavior independently of the tone-sucrose associations.

This is an excellent point. We have carefully analyzed the effects of silencing MS GABAergic neurons, MS SOM neurons or MS SOM projections to LHb on licking behavior during the sucrose exposure. As summarized in **Extended Data Figure 6**, these silencing experiments significantly reduced the licking rate to sucrose as compared to GFP control mice. These results suggest that silencing of the MS_{SOM}→LHb pathway indeed reduces the motivation of animals to approach the reward under our experimental conditions. Nevertheless, silencing of this pathway did not completely abolish the reward approaching behavior, suggesting that there are other reward pathways also mediating this behavior (page 17).

3. Fig. 2 a-p shows that inactivation of PBN and NST decreases MS neuronal responses to sucrose, whereas inactivation of MS enhances lat habenula responses to sucrose. There is a missed opportunity here to link these activity changes to behavior (sucrose licking).

Similar to our reply to point 2, we have now provided lick rate data before and after muscimol application in NST, PBN and LHb (**Extended Data Figure 4**). It became clear that the lick rate to sucrose was reduced after silencing NST, PBN and LHb, whereas the infusion of vehicle saline had no effects. These data have provided a link between activity changes and behavioral changes, further supporting the involvement of the examined neural pathway in the reward behavior.

4. Fig. 3 See point 2.

See our reply to point 2.

5. Fig. 4 – electrophysiological responses in MS are similarly elevated both during tone and sucrose. In the fiber photometry experiments responses to sucrose are much stronger than responses to tone. How is this discrepancy explained and what does it mean? Fig 5: same issue as point 5.

We would like to apologize for the confusion. Both mentioned data were from electrophysiological recording experiments. The plot shown in the original Figure 4f was from one animal. In the revised figure (now **Figure 5f**), we have replaced the plot by averaging traces across multiple animals. We have observed that the responses of Vgat+ neurons to the reward predicative cue could be highly variable, which might be largely attributed to individual differences in learning efficiency. Nevertheless, despite the variations in the measured plasticity level, consistent increases in CS tone responses were observed for both Vgat+ and SOM+ populations in MS.

7. Fig. 6: If SOM is not required for expression of learned responses (just for encoding), how is it related for innate responses to sucrose if at all?

As supported by our new analysis results (**Extended Data Figure 6b**), MS SOM neurons are involved in the innate responses to sucrose. By relaying the reward (i.e. unconditioned stimulus) information, they are required for the formation of appetitive association (i.e. induction). However, after the induction, the reward memory (and likely memory retrieval) may be mediated by other circuits so that the expression of learned behavior does not depend on MS SOM neurons anymore. We have provided such explanation in the revised manuscript (page 20). We have also cited some previous work showing a separation of induction and expression of conditioning effects (page 20).

8. Fig 7: same as before: is MS input to lat habenula involved only in reward associations or also in other aspects of reward.

As shown by our new analysis results (**Extended Data Figure 6c**), the MS_{SOM}→LHb pathway is not only involved in reward associative learning but also in the “innate” reward behavior, as silencing of the projection decreased licking to sucrose.

In summary, the most confusing parts of the study relate to the distinction of MS SOM neuron involvement in innate vs learned (associated) taste reward and in the differences in data on MS neuronal activity collected with different methods. Also, this reviewer subscribes to the view that stimuli are not intrinsically rewarding (see work of Kent Berridge) but only when assigned positive value during processing in the brain. This is especially true when studies are carried out under water deprivation conditions, as it was done here.

Thanks to the reviewer for these constructive comments. We do agree with the reviewer that stimuli cannot be intrinsically rewarding but are only rewarding when assigned a positive value. In the revised the manuscript, we have removed phrases of “intrinsically” or “innately” rewarding and interpret our results as MS SOM neurons encoding reward signals and required for the formation of reward association. We have cited Berridge’s work.

REVIEWERS' COMMENTS

Reviewer #2 (Remarks to the Author):

The authors have substantially addressed my questions. I do not have other concerns.

Reviewer #3 (Remarks to the Author):

The authors have provided new experimental data and clarifications on the previously raised issues. These additions adequately address the concerns and strengthen the paper. Overall, this is an important contribution to our understanding of the circuit organization of reward behavior.

A brief summary of the main findings (250 characters including spaces)

Reward behavior such as food seeking is essential for survival. Shen et al. discovered an ascending neural pathway mediating transformation of rewarding taste signals and reward-cue associative learning via somatostatin neurons in the medial septum.